



# Hyper-resolution PCR-GLOBWB: opportunities and challenges of refining model spatial resolution to 1 km over the European continent

Jannis M. Hoch[1,2,3], Edwin H. Sutanudjaja[1,2], Niko Wanders[1], Rens L.P.H. van Beek[1], Marc F.P. Bierkens[1,4]

[1]Department of Physical Geography, Utrecht University, Utrecht, the Netherlands
[2]Department of Finance, Imperial College London, London, United Kingdom
[3]Fathom, Bristol, United Kingdom
[4]Deltares, Utrecht, the Netherlands

*Correspondence to*: Jannis M. Hoch (*j.m.hoch@uu.nl*)

**Abstract.** The quest for hydrological hyper-resolution modelling is already on-going for more than a decade. While global hydrological models (GHMs) have seen a reduction in grid size, thus far they never have been consistently applied at hyper-resolution (<= 1km) at the large scale. Here, we present the first application of the GHM PCR-GLOBWB at 1 km over Europe. We thoroughly evaluated simulated discharge, evaporation, soil moisture, and terrestrial water storage anomalies, and subsequently compared results with the 'established' 10 km and 50 km resolutions of PCR-GLOBWB. Subsequently, we could assess the added value of this first hyper-resolution version of PCR-GLOBWB as well as understand model and data requirements for future improvements.

We found that for most variables epistemic uncertainty is still large. Merely for simulated discharge we can confidently state that model output at hyper-resolution improves over coarser resolutions. This first large-scale hyper-resolution modelling attempt shows that applying a GHM consistently is by now feasible with improved data availability and computer power. Also, simulated discharge improves due to better representation of the river network at 1 km. However, currently available observations are not yet widely available at hyper-resolution or lack sufficiently long timeseries, which makes it difficult to assess the performance of the model for other variables at hyper resolution. At the model side, hyper-resolution applications require improved parameterization and implementation of physical processes to be able to resemble the dynamics and spatial heterogeneity at 1 km.

With this first application of PCR-GLOBWB at 1 km, we contribute to meeting the 'grand challenge' of hyper-resolution modelling. As such, it should be seen as a modest milestone on a longer journey towards locally relevant model output which requires a community effort from both model developers and data providers.





## 1. Introduction

Climate change is projected to impact patterns and intensity of precipitation and temperature world-wide, resulting in increased occurrence of hydrological extremes and, in turn, to rising hazard probability and risk globally (Dottori et al., 2018; Hirabayashi et al., 2013; Winsemius et al., 2016; Ward et al., 2020b; van der Wiel et al., 2019). To explore large-scale climate adaption options and mitigation measures, as well as the efficacy of risk management plans, output from global hydrological models (GHMs) is increasingly employed as a basis for discussion and policy-making. Examples are the Aqueduct Flood Analyzer (Aqueduct Global Flood Analyzer, 2021; Ward et al., 2020a), the Aqueduct Water Risk Atlas (Hofste et al., 2019), the WWF Water Risk Filter (WWF, 2022), and water2invest (Straatsma et al., 2020; water2invest web service, 2021).

One of the main advantages of GHMs is that their outputs are readily available and provide spatially and temporally consistent estimates of hazard probability and risks (Bierkens, 2015). However, global climate change impacts are often evaluated at the level of river basins or at country-scale. Therefore, it is critical that the use of GHMs, which typically employ a relatively coarse spatial resolution ($\geq$ 5 arc-min, equivalent to around 100 km$^2$ at the Equator), is fit for purpose and the current GHMs are used as intended (Ward et al., 2015). Nevertheless, the actual impact of hydrologic hazards happens at the local level and may vary greatly within countries which pushes the boundaries of applicability of the current generation of GHMs. To capture these important spatial variations and to eventually make GHMs "locally relevant" (Bierkens et al., 2015) and usable by local water managers (Beven and Cloke, 2012), it is hence the current understanding that models need to move towards "hyper-resolution" (Wood et al., 2011; Bierkens et al., 2015), that is to a spatial resolution below 1 km$^2$ (30 arc-seconds).

Global models have their own *raison d'etre* as discussed in Ward et al. (2015), for instance analyses over data-sparse areas or continuous trans-boundary simulations, thereby complementing bespoke local and country-scale models which may already be able to run at hyper-resolution. Moving towards hyper-resolution may potentially improve these analyses and, as a result, their meaningfulness and applicability. However, due to issues with, *inter alia*, model parameterization as well as computational demand, running GHMs at a meaningful and actionable spatial resolution over large extents was considered to be a "grand challenge" (Bierkens et al., 2015; Wood et al., 2011).

With computational power increasing and input data sets becoming available at ever finer spatial resolutions, the spatial resolution of current GHMs has increased in past years too. Currently, most GHMs can be run globally at a resolution of 5 to 6 arc-min according to Bierkens et al. (2015). However, a grid cell at a resolution of 5 arc-min still translates to roughly 10$^2$ km$^2$ at the Equator. Recently, various modeling attempts were made to further increase spatial resolution or to pave the way towards it. For example, the HydroBlocks model (Chaney et al., 2016) was coupled with remotely sensed data to obtain soil moisture estimates at 30 m spatial resolution over the continental United States of America (CONUS) (Vergopolan et al., 2020, 2021). While HydroBlocks brings the simulations to an effective 30 m resolution, it still takes advantages of grouping regions based on hydrological similarity to reduce computational demands. Maxwell et al. (2015) and more recently O'Neill et al. (2021) applied and evaluated the hydrological model ParFlow configured over the CONUS at 1 km spatial resolution (named PFCONUS), constituting the first-of-its-kind studies covering such an extensive area at 'true' hyper-resolution. However, PFCONUS is not considered to be a GHM as its set-up is more specifically tailored towards applications over CONUS than it would be the case for GHMs. Aerts et al. (2021) assessed changes of model accuracy of the hydrological model wflow_sbm (Schellekens et al., 2020) at multiple resolutions below 1 km$^2$ for the CAMELS dataset (Newman et al., 2015; Addor et al., 2017) again solely covering the CONUS. While the spatial resolution applied is 'hyper-resolution', the lack of spatially-continuous simulations over large extents make it challenging to transfer findings to the realm of GHMs. Other studies already proposed scalable routing networks (Thober et al., 2019) and measures how to seamlessly model across scales (Samaniego et al., 2010, 2017) which are essential when moving towards hyper-resolution.

On a more general note, the call for "hyper-resolution" is strongly driven by an expected increase of model accuracy and consequently applicability. One hypothesis thereby is that particularly locations with a small upstream area will profit as more interactions will be simulated there and that representation of the storage and



response times in these regions are improved. The question whether this is really the case depends also on the ability to populate the many fine resolution cells with accurate parameters (Wood et al., 2011; Beven and Cloke, 2012). It is believed that this is not possible without additional locally specific information and as such Beven et al. (2015) concluded that hyper-resolution modelling will not achieve the necessary accuracy. Thus, we first need to establish an understanding to what extent accuracy of GHMs is improved when only their resolution is increased while still employing globally available datasets currently used for parameterizing GHMs.

A second avenue towards improved model accuracy is the use of more accurate meteorological forcing. In studies using GHMs, it is shown that particularly the quality and spatial resolution of meteorological variables, notably precipitation, can have a pronounced impact on model accuracy (Beck et al., 2020; Towner et al., 2019; Biemans et al., 2009). Hence, a second question is raised whether it is needed to go all the way to "model hyper-resolution" or if using "forcing hyper-resolution" with current GHMs already suffices. This would reduce the need for updated model parameterization and design as well as greatly shorten run times and reduce model data storage requirements. A locally relevant spatial resolution could then be achieved using post-processing tools, for instance.

To answers test these hypotheses, we developed a 30 arc-seconds (~1 km) version of the existing 10 km PCR-GLOBWB model (Sutanudjaja et al., 2018) for the entire European continent, constituting the first GHM at hyper-resolution. We use this model to test how a stepwise increase in spatial resolution (30 arc-min/~50 km, 5 arc-min/~10 km, 30 arc-sec/~1 km) impacts model accuracy. To test the impact of forcing resolution, the model simulations at the different spatial resolutions were forced with ERA5-Land data (Muñoz-Sabater et al., 2021) at the same spatial resolutions or coarser. Finally, we combine both experiments to disentangle the relative importance of the model and forcing resolution. Except for the precipitation forcing, all simulations make use of the same input and parameter sets and are downscaled or upscaled only if necessary. In our analysis, we focus not merely on simulated discharge as done by Aerts et al. (2021), for instance, but also on terrestrial water storage anomalies, evaporation, and soil moisture to provide a more holistic picture of model performance than what is typically the norm (see O'Neill et al., 2021). Such a very first thorough analysis of a hyper-resolution GHM is in the first place not intended to claim that hyper-resolution should be the 'new normal' for GHMs, but rather to gather opportunities and challenges, thereby helping to identify the most effective model development pathways and advancing current efforts to produce spatially consistent and locally relevant estimates at (sub-)kilometer scale.

With hydrologic hazards already having increased and projected to increase further (see e.g. Winsemius et al. (2016), Alfieri et al. (2017), Samaniego et al. (2018), van der Wiel et al. (2019), and the work of the World Weather Attribution initiative (Kew et al., 2021; Philip et al., 2019)), it is paramount to better understand their impact not only for entire economies, but also more granular on the sub-national to community level. Once actionable local model output becomes available, large-scale *and* target-oriented adaptation and mitigation measures can be devised. In anticipation of the first global-scale 1 km GHM, the here presented evaluation of the relative impact of refining the spatial resolution of the model and its forcing may provide an important contribution to the discussion and advancement of global hydrologic modelling at hyper-resolution.

## 2. Methods, data, and study area

### 2.1. Study area and input data

In addition to the already existing global schematizations of PCR-GLOBWB at 30 arc-min (roughly 50 km cell length at the Equator) and 5 arc-minutes (roughly 10 km cell length at the Equator), respectively, we developed a novel 30 arc-seconds (roughly 1 km cell length at the Equator) schematization. Even though the 1 km model schematization of PCR-GLOBWB can be scaled up globally, we selected Europe to test the model and assess output due to the quality of input data (soil, geology, meteorological forcing), the abundance of observational validation data, particularly GRDC discharge observation data. Finally, it will bring the added benefit of reduced computational demand and storage allocation, which is challenging for hyper-resolution continental scale applications. The study period for which each run was executed is 1981 until 2019 at a daily timestep. However, to reduce storage requirements and due to the timestep of most observations, our evaluations focused on the monthly timestep except for discharge analysis where both daily and monthly timestep were used.



Each of the model resolutions[1] was forced with precipitation at the model resolution or coarser, yielding in total six combinations of model and forcing spatial resolutions (see Table 1).

**Table 1: Overview of model and forcing as well as used run names**

| Model | Forcing | Run name |
|---|---|---|
| PCR-GLOBWB 50 km | ERA5-Land 50 km | 50k_50k |
| PCR-GLOBWB 10 km | ERA5-Land 50 km | 10k_50k |
| PCR-GLOBWB 10 km | ERA5-Land 10 km | 10k_10k |
| PCR-GLOBWB 1 km | ERA5-Land 50 km | 1k_50k |
| PCR-GLOBWB 1 km | ERA5-Land 10 km | 1k_10k |
| PCR-GLOBWB 1 km | ERA5-Land 1 km | 1k_1k |

For all three spatial resolutions, identical model input data and parameter sets are used to guarantee comparability across set-ups. The 'default' resolution is 1 km, meaning that all model development including runoff, soil, and groundwater processes, was geared towards this resolution and all other resolutions are derivatives. A detailed overview over datasets used for model parameterization and forcing input can be found in A1. Appendix A, including the scaling techniques used to derive them at various resolutions (i.e., 1 km, 30 km and 50 km). Each

run includes water demand and use for irrigation, industry, livestock, and household as well as accounts for reservoir storage based on the GranD database (Lehner et al., 2011). Note that none of the runs were calibrated to not have differences in parameterization affect the comparability across different model resolutions and forcing resolutions.

### 2.2. Model evaluation

To fully grasp the impact of an increase in spatial resolution on hydrological states and fluxes, various model output variables were compared to observations: simulated discharge, total evaporation, terrestrial water storage anomaly, and upper layer soil moisture. Additionally, we benchmarked the computational demand of the various runs performed. While discharge was evaluated at locations, the other variables were evaluated at the water province level (Straatsma et al., 2020). Hereby water provinces smaller than the coarsest cell size of the

observational data used (here: the original 3-degree footprint of the GRACE/GRACE-FO data, see section 2.2.4) were merged with the adjacent province having the longest common border.

It should be noted that by evaluating the 1 km hyper-resolution outputs at the water province level, we can only assess the overall performance of these simulations, rather than site-specific improvements. While it would be preferable to evaluate the model simulations at the native 1 km resolution this is difficult due to the coarse spatial

resolution of some of the validation datasets.

### 2.2.1. Simulated discharge

To evaluate simulated monthly discharge, we employed data from the Global Run-off Data Centre (GRDC). To assess the accuracy of monthly simulated discharge, we determined the Kling-Gupta Efficiency (KGE; Gupta et al. (2009)). Based on Knoben et al. (2019), a value of KGE ≥ -0.41 indicates that the model improves upon the

mean flow benchmark.

For meaningful analyses, we used only GRDC stations from our dataset which met the following requirements: *i)* the timeseries of a station extends at least until 1991; *ii)* the timeseries of a station contains at least 10 years of data; *iii)* the catchment area of a station is at least 400 km$^2$, which is equivalent to four upstream cells for the 10k resolution. This yielded in total 564 stations. Since this large number of stations did not allow manual matching

of their locations and the model river network, we determined the corresponding model cell by searching for the smallest deviation between mean observed and simulated discharge in proximity of the location as specified by

---

[1] The term 'resolution' refers to spatial resolution throughout this manuscript, if not specifically stated otherwise





GRDC. This radius was determined by testing, and we consider it a balanced distance between sufficient search space and allocating discharge stations over non-realistically large distances.

For all runs, we evaluated the KGE distribution as is and additionally compared KGE values as function of upstream catchment area per station. This way we expect to obtain a better picture whether finer spatial resolution has a more marked impact on upstream stations compared to downstream stations. To analyse this, we categorized all stations depending on their catchment area with a minimum catchment area of 400 km$^2$ due to the above-described selection criteria. Stations with a catchment area falling in the 25 % quantile were considered upstream stations, those falling in the 75 % quantile downstream stations, and all remaining ones midstream stations. One hypothesis is that a finer model spatial resolution improves especially discharge simulations in upstream areas due to the refined channel network and thus better representation of upstream processes.

Per station, the change of obtained KGE between two different runs can be compared by computing the KGE skill score ($KGE_{ss}$) based on the subsequent equation (Towner et al., 2019):

$$KGE_{ss} = \frac{KGE_a - KGE_{ref}}{1 - KGE_{ref}}$$

Here, $KGE_a$ is the KGE of the model run under consideration, while $KGE_{ref}$ is the KGE of the reference run for which we use the 50k_50k run. Positive $KGE_{SS}$ indicates improved skill, whilst a negative score represents a decrease in skill.

Thus far, we focused on monthly discharge results only as it reduces file size and makes the required analyses manageable. Moreover, the runs did use the simpler routing scheme implemented in PCR-GLOBWB, named 'accuTravelTime', which has limitations on accurately simulating discharge since it relies on a characteristic method and fixed flow velocities. For daily discharge simulations, particularly in larger channels, a kinematic or dynamic wave approximation is needed. We did not apply these as it would lead to very large computation times at 1 km resolution under the current PCR-GLOBWB code. Nevertheless, we think it may add to the evaluation if we also assessed how fast model processes, for which a short temporal resolution is crucial, profit from an increased spatial resolution of the model, as argued by Melsen et al. (2016). To that end, we compared obtained KGE values obtained with monthly output per catchment area category.

### 2.2.2. Upper soil moisture

For validating the average soil moisture of the upper layer, we compare simulations with remotely sensed ESA-CCI soil moisture data v06.1 (Dorigo et al., 2017; Gruber et al., 2019). The native temporal resolution is daily and the spatial resolution is 0.25 degree (roughly 25km at the Equator). It is noteworthy that the observed data is representative for the upper 5 cm of the soil column whereas simulated upper soil moisture from PCR-GLOBWB is an average of the first 30 cm. It should additionally be mentioned that an older version of ESA-CCI soil moisture data is one of the input data for GLEAM evaporation data (see section 2.2.3). Nevertheless, we can still use the EDA-CCI data as indicator for actual soil moisture dynamics. By computing the spatial mean per water province per time step for both observation and simulation, the resulting timeseries were used to derive the relative root-square-mean-error (RRMSE) per water with the subsequent equation:

$$RRMSE = \frac{RMSE(obs, \; sim)}{\sigma(obs)}$$

Here, RMSE(obs, sim) is the root-square-mean-error between observation and simulation timeseries, and σ(obs) the standard deviation of the observation timeseries.

### 2.2.3. Total evaporation

Evaporation makes up a significant factor in in the terrestrial water cycle as well as water balance, strongly determining water availability. Accurate simulation of evaporation is therefore paramount. Unfortunately, there is only little independent observation data available with sufficient temporal and spatial variation and extent.





Therefore, to benchmark simulated total evaporation per water province, we used data from the remote sensing-based dataset GLEAM version 3.5a (Global Land Evaporation Amsterdam Model; Gonzalez Miralles et al., (2011), Martens et al., (2017)), covering the period from 1980 to 2020. GLEAM v3.5a has a native monthly temporal resolution and a spatial resolution of 0.25 degree (~25 km). Since the algorithms implemented in
GLEAM aim much more on correctly simulated correct evaporation than PCR-GLOBWB, and because GLEAM employs different (remotely sensed) input data sets, the choice to use a partly model-based benchmark is defensible. Similar as for upper soil moisture evaluation, the RRMSE was determined per water province.

### 2.2.4. Terrestrial water storage anomaly

By validating model total water storage against remotely sensed terrestrial water storage, it is possible to evaluate
the model's storage dynamics. As such, it provides a more holistic validation including the entire hydrological system state.

Here, simulated terrestrial water storage anomaly was validated against JPL-Tellus GRACE/GRACE-FO (Kornfeld et al., 2019) data for the period 2002 until 2019. While the native spatial resolution of the data is at 3 degrees, the data used here is at 0.5 degree. Again, monthly averages were computed for both observation and
simulations to obtain the RRMSE per water province.

### 3.    Results and discussion

### 3.1.    Simulated discharge

We see that simulated discharge, as expressed by the KGE value averaged over all GRDC stations per water province, improves greatly with finer model spatial resolution (Figure 1, Figure 2; Table 3). While the 50k_50k
run yields a negative median KGE values, this picture improves for finer resolutions with the 1k_1k and 1k_10k runs comfortably exceeding the accuracy threshold of -0.41 as defined by Knoben et al. (2019). Higher skill with finer resolution is in line with previous research (Altenau et al., 2017), although this is should not be expected to be the case for all stations due to locality effects in the scaling process (Aerts et al., 2021). As in our case only around 7 % of all stations do not show an improvement at all but roughly 70 % of the stations receive their highest
KGE values for the 1k_1k run (Figure B1), results nevertheless indicate a robust relation between model resolution and accuracy of simulated discharge.





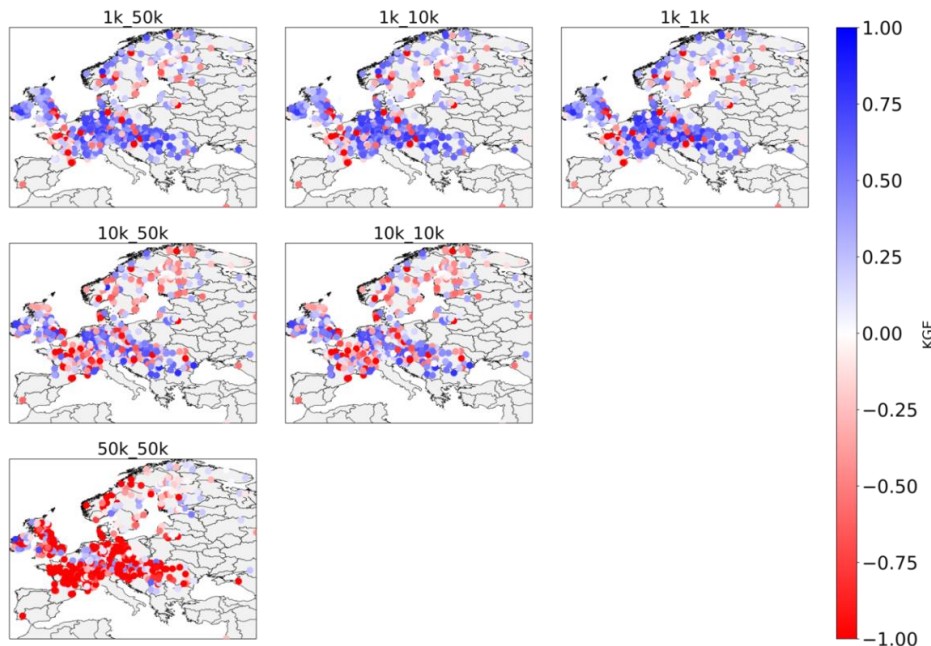

**Figure 1: KGE values of all selected GRDC stations in Europe.**

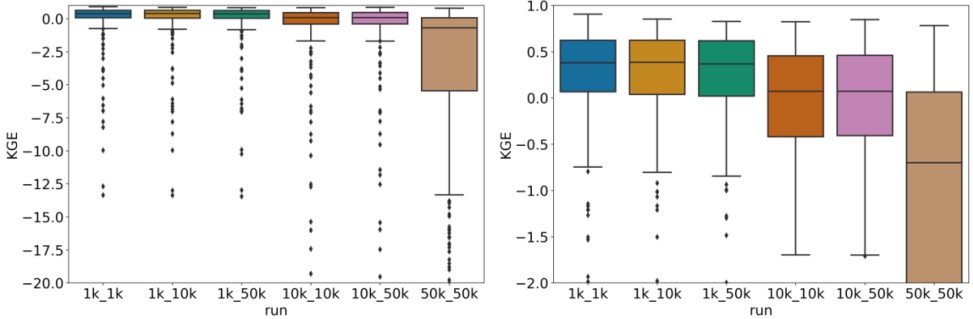

**Figure 2: Boxplot of KGE values obtained for all runs; right zoomed in to smaller value range.**

Results furthermore indicate that the impact of forcing spatial resolution is present but limited to the 1k runs, and even there only the distribution of KGE values contains slightly higher values (as expressed by the whisker extents in Figure 2) while mean KGE values are close to identical. Overall, results suggest that model resolution is a more important determinant of the accuracy of simulated discharge. Due to the minor differences found across forcing resolutions, the subsequent discharge analyses focus on results from the 1k_1k, 10k_10k, and 50k_50k runs.

When analysing results per catchment area category, they also show that KGE values increase when moving to 1 km hyper-resolution, with the magnitude of improvement depending on upstream area (Figure 3). In line with previous results and our hypothesis, employing finer resolutions than the reference 50 km has the most pronounced effect at upstream locations where the 50 km model is seemingly less capable of reproducing observed discharge. For midstream and downstream stations, $KGE_{ss}$ values show the greatest improvement for the 1k_1k run, and

hence a clear beneficial effect of hyper-resolution hydrological modelling. In fact, simulated discharge at 1 km improves over either 10 km or 50 km or both at around 93 % of the GRDC stations assessed (see Figure B1). In




addition to our hypothesis that hyper-resolution is beneficial in upstream areas, these results suggest that a better representation of smaller streams contributes to more accurate discharge simulations at large.

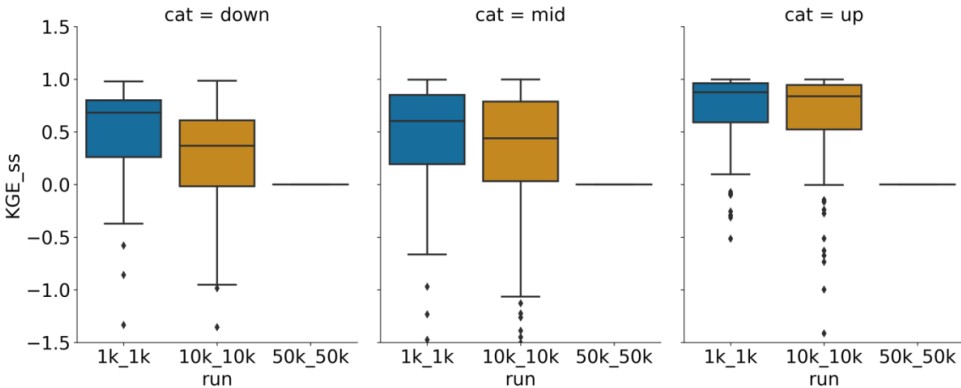

**Figure 3: Boxplots of obtained KGE$_{ss}$ values for all stations categorized as downstream, midstream, and upstream for selected runs.**

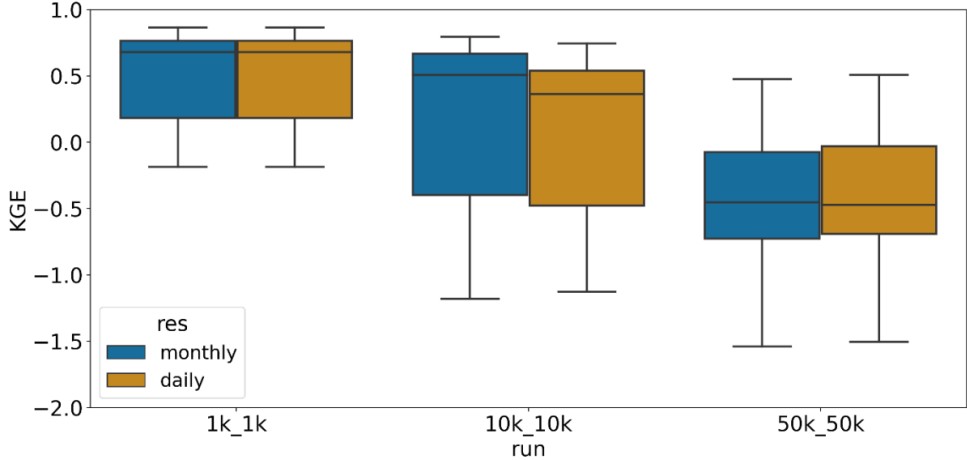

**Figure 4: Comparison between KGE distribution presented as boxplots for different spatial resolution and temporal resolution of discharge output.**

Another hypothesis we tested is whether the accuracy of fast processes simulated at daily resolution, such as discharge generation, improve compared to longer aggregation periods, for example months, with finer spatial resolution. This is in line with Melsen et al. (2016) who argues that 'the calibration and validation time interval should keep pace with the increase in spatial resolution'. Based on Figure 4, this hypothesis cannot be confirmed for the PCR-GLOBWB model. We do, however, want to stress that these results do not mean that moving to hyper-resolution does not yield improvements for shorter temporal intervals at all. It is highly likely that the simplistic routing schematization used in this experiment limits the improvement of daily discharge dynamics. Using more advanced routing schemes was not feasible due to the resulting high computation demand. It can thus be assumed that the here presented daily discharge estimates do not fully represent the potential benefits which could be achieved with hydrological simulations at hyper-resolution.



## 3.2. Upper soil moisture

To assess the model skill in reproducing upper soil moisture, we analysed RRMSE values obtained per water province (Figure 5). Overall, we find areas with consistently low and high RRMSE values. Particularly the UK, Ireland, southern Sweden and Norway, the Rhine-Meuse delta, and a larger area centred around Austria have high RRMSE values. Large parts of Spain, Portugal, and Finland as well as northern Sweden show overall good model performance as indicated by low RRMSE values.

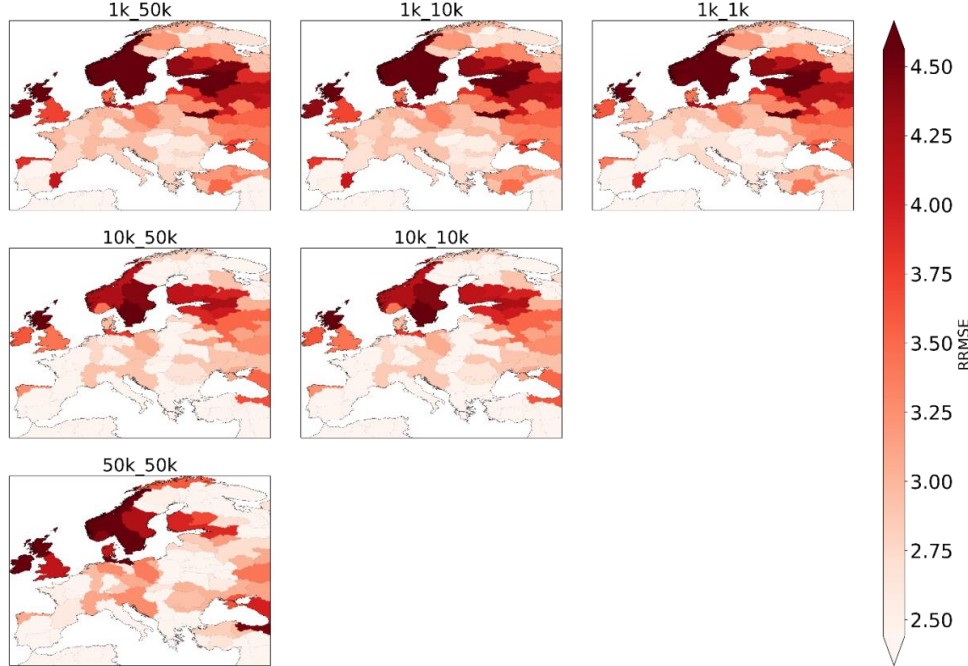

**Figure 5: Relative Root Mean Square Error (RRMSE) values computed per water province based on simulated upper soil moisture and observations from ESA-CCI.**

Comparing RRMSE values across both model resolutions (Figure 6A and Table 3) reveals that the 10k runs, when benchmarked against the 1k run, produce overall lower RRMSE values and are thus more accurate, although differences between median values are overall small especially between the 1k_1k and 10k_10k runs. It is in line with our expectations that the 50k run has the highest RRMSE values. Figure 6B and Figure 6C compare RRMSE values across forcing resolutions, indicating that employing coarser forcing resolution decreases model accuracy for the 1k runs while not having significant impact on the 10k run (see also Table 3) – a similar pattern to that found in the discharge evaluation.

It should be noted, however, that observed and simulated soil moisture values are based on different soil depths, which may have a not further quantifiable impact on analysis results. The fact that a similar pattern is found when assessing timeseries variability as expressed by the coefficient of determination $R^2$ indicates that the relation between observation and simulations is well-represented also when assessing RRMSE values (Figure B2).





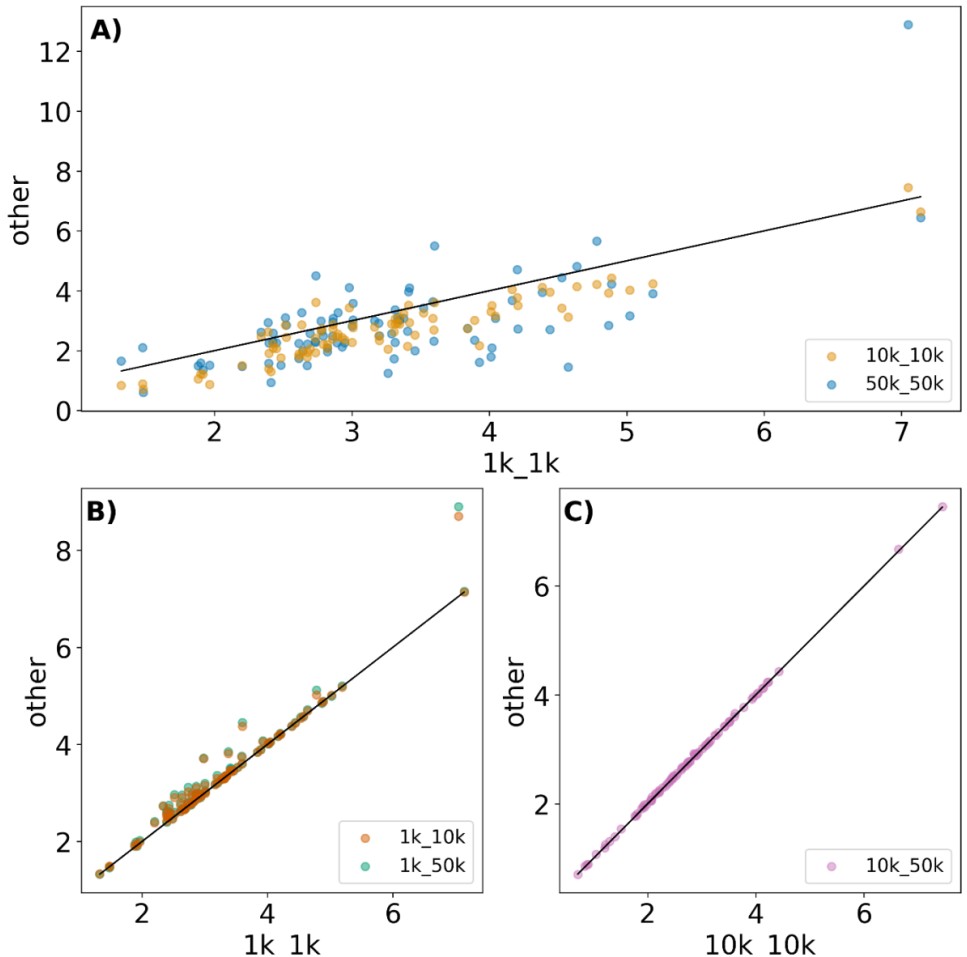

**Figure 6: Scatterplot of RRMSE values obtained in evaluation of simulated upper soil moisture with ESA-CCI data for: A) comparing different model resolutions; B) comparing different forcing resolutions for 1k runs; C) comparing different forcing resolutions for 10k runs.**

Ranking the accuracy of the runs by their mean RRMSE value, we see that the 10k runs perform best, followed by the 1k_1k run (Table 3). The fact that the 10k_10k runs clearly outperform the 1k_10k and 1k_50k runs indicates that a too coarse forcing resolution can eradicate potential benefits of finer model resolution, and that employing a coarser model resolution with a (relatively) fine forcing resolution may lead to similar accuracy.

### 3.3. Total evaporation

Results indicate that model accuracy is lowest for the 1k runs, followed by the 50k run (Figure 7, Figure B3A; Table 3). The 10k runs show overall best performance in resembling evaporation estimates from GLEAM. We furthermore find that RRMSE values are smaller and range between 0.22 and 1.78, indicating overall better agreement between simulation and observation than for surface soil moisture. The deterioration of model accuracy at 1 km scale is visible over the entire European continent except for Great Britain where we see improvement
when moving towards 1 km model resolution.




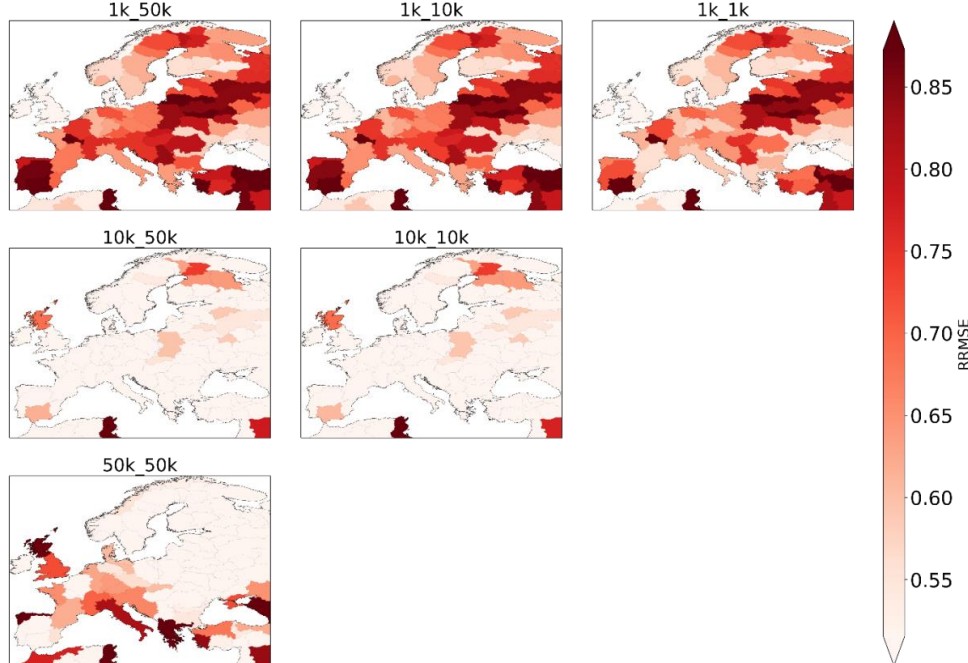

**Figure 7: RRMSE values computed per water province based on simulated total evaporation and evaporation data from GLEAM.**

The reason for this deviation from the positive performance changes observed for discharge simulations is currently unknown. We see three possible reasons. First, the runs with 1 km model resolution are not able to reproduce evaporation dynamics, possibly as a result of epistemic uncertainty surrounding soil and vegetation parameterization at that spatial scale. Second, GLEAM, the dataset employed for model evaluation, tends to be favourable for coarser model resolutions, possibly as result of its own coarse spatial resolution of 0.25 degree which might lead to a favourable comparison at coarser spatial resolutions. Or third, that the algorithms of the GLEAM model at 0.25-degree resolution cannot fully, as carefully developed as they are, represent local values.

What is consistent with the soil moisture evaluation, however, is that employing a coarser forcing resolution increases RRMSE values for the 1k runs, and hence decreases model accuracy, but has no impact on the 10k runs (Figure B3B, Figure B3C; Table 3).

### 3.4. Terrestrial water storage anomalies

As with all previous evaluations, evaluating terrestrial water storage (TWS) anomalies indicates that coarser forcing resolutions yield reduced model accuracy, but in this case the impact is smaller than for previously assessed variables (Figure B4B, Figure B4C; Table 3).

Figure 8 shows that there are distinct hotspots (Scandinavia, the UK and Ireland, Italy, and water provinces along the Atlantic and Mediterranean coast) where model accuracy is limited, regardless the model resolution or forcing resolution employed. Furthermore, the same pattern occurs as with total evaporation, namely that overall RMSE values are lowest for model resolutions coarser than 1 km, in that case the 50k run (Figure B4A; Table 3). These findings, again, may be related to the fact that the large original footprint of the GRACE data of 3 arc-min or roughly 300 km cell length, despite downscaling it to 0.5 degree, favours coarser resolutions. In fact, the match in spatial resolution may be explaining some of the good performance of the 50 km run. Consequently, the (potentially) added value of finer model resolutions may not be captured when using GRACE data for evaluation.





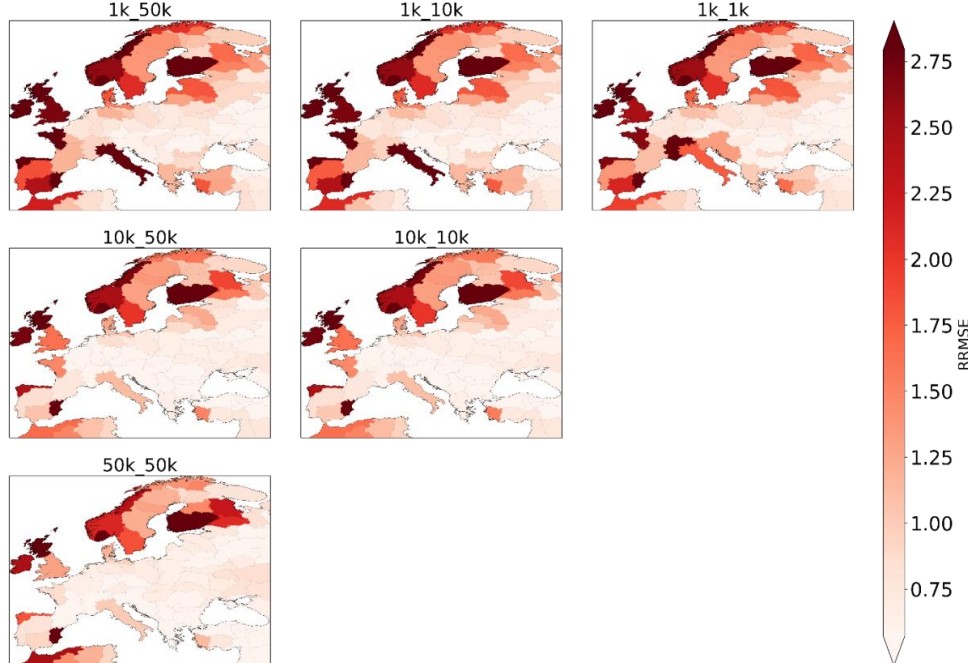

**Figure 8: RRMSE values computed per water province based on simulated total water storage anomalies and terrestrial water storage anomalies from GRACE/GRACE-FO.**

Another possibility is that parameterization for the groundwater response times (the 'j-parameter' in PCR-GLOBWB), which has a major impact on groundwater storage dynamics, needs to be adjusted for finer spatial resolutions as it does not (yet) correctly scale with the underlying drainage network. Consequently, model response, particularly with respect to regional groundwater dynamics, may be too slow at model resolutions finer than the thus far default 10 km resolution, and thus the groundwater response is out-of-sync with observations. A lower correlation (expressed here as the coefficient of determination $R^2$) may thus result in high RRMSE values. To further analyse this, we categorized each water province according to its $R^2$ and RRMSE value (Figure 9). Indeed, results suggest that $R^2$ is an overall good predictor of RRMSE: water provinces with a high $R^2$ tend to have low RRMSE values and vice versa, accounting for in total 66 % to 68 % of all provinces. Consequently, issues related to correct scaling of the recession parameter may indeed contribute to the not-improvement at finer model resolutions.



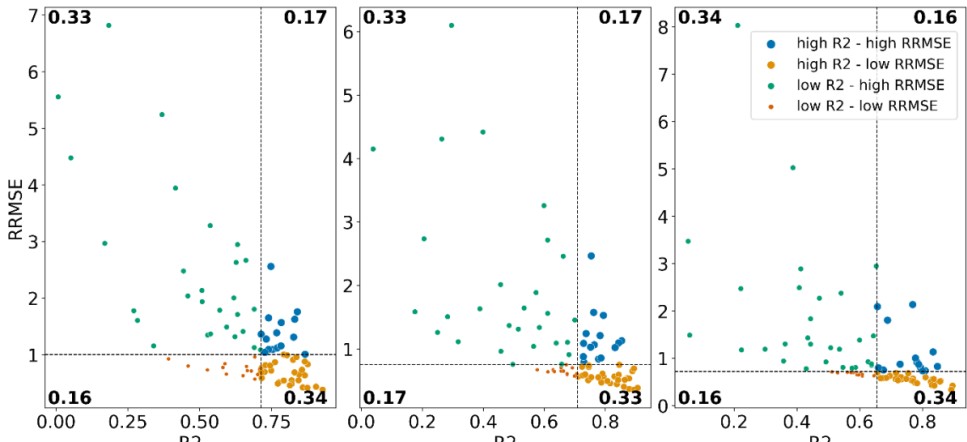

**Figure 9: Plot of R2 and RRMSE values per water province, categorized in four quartiles according to their R2 and RRMSE values, respectively, for 1k_1k, 10k_10k, and 50k_50k runs (from left to right). Values in corners of quartiles represent the fraction of data points in the respective quartiles.**

For additional underpinning of the drivers of limited model skill to reproduce TWS anomalies, we selected two provinces (Figure 10; for their locations see Figure B5): one from the 'low R2 – high RRMSE' category (ID: 204; R2: 0.184, RRMSE: 3.515) and one from the 'high R2 – low RRMSE' category (ID: 224; R2: 0.928, RRMSE: 0.382). This figure not only confirms the assumption that a high $R^2$ can explain high RRMSE values, and as such shows that the 1 km version of PCR-GLOBWB can accurately reproduce dynamics of TWS anomalies for a third

of the water provinces (Figure 9). It, however, also points to another issue at 1 km model resolution, namely a faster decline of TWS at 1 km resolution.

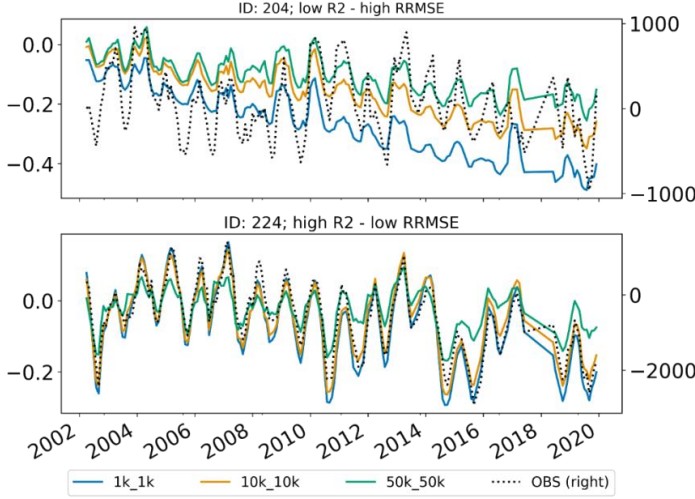

**Figure 10: Timeseries of TWS anomalies for PCR-GLOBWB at various resolutions and as observed by GRACE/GRACE-FO for two water provinces. Note the GRACE/GRACE-FO timeseries refers to the secondary y-axis at the right.**

Our hypothesis for this behaviour is currently that at hyper-resolution the greater spatial heterogeneity leads to locally lower soil moisture and in turn more irrigation demand and thus groundwater depletion, while at coarser resolution these soil moisture gradients would be averaged out in space and thus reducing the need for irrigation.





This would be supported with the location of water province with ID 224 which is along the Spanish Mediterranean coast where groundwater irrigation is dominant.

A third driver could be the accumulation of water storage over time, particularly in snow-dominated areas where the downscaled temperature at 1 km is mostly below the freezing point. In these instances, PCR-GLOBWB does not yet contain enough physical processes to re-distribute snow to other 1 km cells (e.g., glaciers or avalanches). Indeed, correcting for snow cover of the 1k_1k run had effect when evaluating TWS anomalies (Table 2) for two water provinces (not surprisingly located in and around the Alps, see Figure B5), suggesting that the current processing of snow cover may limit model accuracy locally for hyper-resolution runs.

**Table 2: RRMSE values for two selected water provinces without and with correction for snow cover when evaluating simulated TWS anomalies.**

|  | RRMSE (w/o correction) | RRMSE (w correction) | RRMSE reduction |
|---|---|---|---|
| ID 45 | 1.318 | 0.597 | 45 % |
| ID 179 | 5.553 | 0.862 | 16 % |

**Table 3: Overview of median evaluation metrics per run per evaluated variable except for discharge.**

| dataset | metric | 1k_1k | 1k_10k | 1k_50k | 10k_10k | 10k_50k | 50k_50k |
|---|---|---|---|---|---|---|---|
| GRDC | median KGE | 0.38 | 0.38 | 0.37 | 0.07 | 0.07 | -0.70 |
| ESA-CCI | median RRMSE | 3.00 | 3.13 | 3.16 | 2.67 | 2.68 | 2.63 |
|  | median R2 | 0.30 | 0.30 | 0.30 | 0.32 | 0.32 | 0.28 |
| GLEAM | median RRMSE | 0.66 | 0.70 | 0.70 | 0.40 | 0.40 | 0.46 |
|  | median R2 | 0.97 | 0.97 | 0.97 | 0.97 | 0.97 | 0.98 |
| GRACE | median RRMSE | 1.00 | 0.99 | 1.00 | 0.75 | 0.75 | 0.72 |
|  | median R2 | 0.71 | 0.71 | 0.71 | 0.71 | 0.71 | 0.66 |

### 3.5. Computing and data storage demand

Thus far, we focused on model accuracy, but another important aspect of hyper-resolution hydrological modelling is the computational demand. Table 4 shows computational demands for different model resolutions using the Dutch super-computer Snellius. Here, the run times and disk storages required for different resolutions were compared for performing the European continental-extent runs over three decades (1981-2019).

Without parallelization, the wall clock time for a 10 km run was about 36 hours. This entails that the 1 km model, having the size of about 100 times the 10 km model, would result in a wall clock time of 3,600 hours (~150 days). Hence, to speed up computation, the 1 km model was divided into 45 groups of river basins such that it could be run as 45 individual runs (masks). With this simple parallelization technique, the wall clock time for the 1 km model was reduced to about 204 hours (~8.5 days), which was limited by the largest river mask covering the Danube River basin and its neighbouring smaller river basins.

**Table 4: Overview of indicators for computational demand of the runs at different model resolutions. The file sizes refer to output for one variable.**

|  | 1k | 10k | 50k |
|---|---|---|---|
| Run time |  |  |  |
| *- Without parallelization* | 3,600 h | 36 h | 9h |
| *- With parallelization* | 204 h | not parallelized |  |
| File size monthly data | 25.000 MB | 500 MB | 15 MB |
| File size daily data | 750.000 MB | 15.000 MB | 450 MB |

The indicators shown in Table 4 show that performing hyper-resolution simulations is still pushing the boundaries of what is possible, although it is likely that better much better performance is possible by better parallelizing the



code. Nevertheless, without access to high-performance computers with sufficient CPU, memory, and disk space, obtaining results at hyper-resolution at European extent is not realistically feasible.

## 4. Conclusion and recommendations

In this study, we assessed how refining both model and forcing resolution from 50 km to 10 km to hyper-resolution
1 km influences model accuracy. To that end, model grids at these spatial resolutions were combined with forcing data at the same or coarser spatial resolution, and a set of model state and fluxes were evaluated against observations.

Overall, the influence of finer model resolution is less clear, whereas results suggest that employing a fine forcing resolution will benefit the results. Our work shows that too coarse forcing resolution can eradicate benefits of
model hyper-resolution. Hence, if forcing is only available at coarse spatial resolution, applying a hyper-resolution model will not provide any performance gain and only increase computational demand. However, results also show that model resolution is driving overall output accuracy: if model skill at a given model resolution and coarse forcing resolution is low (or already high), employing finer forcing resolution will only yield limited improvement.

While for some simulated discharge moving to hyper-resolution model resolution shows an improved accuracy, such improvement is not visible for internal state variables (surface soil moisture) or even show an opposite tend (evaporation, terrestrial water storage anomaly). A clear answer what causes these ambiguous results is currently, however, not possible as epistemic uncertainty is large: either the observations, the model, the mismatch in spatial resolutions between observations and model, or all of them may lead to mixed results. As this study is at the
forefront of hyper-resolution modelling over Europe, it is not yet possible to benchmark our results against output from comparable models to gain additional insights. Once 1 km output from more models is available, the here presented results should be revisited and put into perspective.

To reduce the uncertainty surrounding the analysis, observational datasets need to move into the realm of hyper-resolution as well. Currently, the difference between the finest resolution of the model (1 km) and of observational
datasets that cover the entire European domain (25 to 50 km) is too large, leading to mismatches in spatial representativeness of the model and observations. As only recent satellite missions, such as ESA Sentinel mission or commercially-driven ones, can produce data with sufficient local detail, it will still take some time until the observed period is sufficiently long for robust model evaluations. For example, surface soil moisture data collected by Sentinel1B started in 2015 only, whereas the here used ESA-CCI data covers the period 1978 to 2020. Vice
versa, the availability of 1 km model output over large extents now offers a great opportunity to make full use of the benefit of these novel data which would have otherwise been too detailed for 10 km or even 50 km model results.

Additionally, the modelling side needs advances to provide clearer answers whether hyper-resolution modelling is of added value. While the here presented hyper-resolution work is not solely the same old model with more grid
cells for better visualization (as discussed by Beven et al. (2015)), but actually populated with as much high-resolution data as possible, there is still room for improvement as the currently used parameterization design is not fully scalable from the current default of 10 km to 1 km. Clearly, some parameters or sub-grid approaches can be replaced by actual data at 1 km. In light of the growing number of openly-available datasets it may also be needed to review some parameter values currently used or to include additional physical processes to capture the
immense natural heterogeneity which can be resembled at the 1 km scale (Clark et al., 2017; Beven and Cloke, 2012). By improving parameterization and implementing more processes at hyper-resolution, we expect that model realism will improve and epistemic uncertainty will decrease.

Efficiently testing various input datasets requires manageable run times. Right now, the run times of 1 km runs are still high, and thus future efforts need to aim at reducing them. Besides using ever more powerful high-
performance computers (HPCs), additional efforts should be taken to improve PCR-GLOBWB's parallelization capacities as implemented in comparable models (Kollet and Maxwell, 2006; Kollet et al., 2010; Bierkens et al., 2015). Additionally, porting Python-based PCR-GLOBWB to a faster programming language like Julia (Bezanson et al., 2017) may yield shorter run times. A recent example of a hydrological model being re-written



in Julia is wflow.jl (van Verseveld et al., 2021). This is of course not to say that all models which are ought to be run at hyper-resolution need to be rewritten, but the question how to deal with the high computational demand of not only running the simulations but also analyzing results – possibly supported by research software engineers (Hut et al., 2017) – is (again) becoming central in meeting the grand challenge of hyper-resolution hydrological modelling.

Shorter run times can have direct positive effects on the selection of the routing scheme. To keep run times within a manageable range, we had to employ the 'accuTravelTime', the largest simplification of the shallow water equations, for this study. Consequently, especially daily discharge simulations are potentially inaccurate. Against this backdrop, the overall accuracy of simulated discharge is beyond our initial expectation , particularly for the hyper-resolution runs with median KGE values of 0.38. Once a speed up of the model is realized, it could be feasible to employ the kinematic wave instead, which already considers friction and water slope, and further improve model accuracy. Alternatively, more advanced routing schemes, including floodplain inundation dynamics, could be included on-demand by model coupling (Hoch et al., 2017, 2019) with hydrodynamic models where 'hyper-resolution' is already achieved (Bates et al., 2018), or by implementing scalable routing schemes (Thober et al., 2019). Based on higher-order simplifications of the shallow water equations, it can be expected that hyper-resolution hydrological modelling significantly improves sub-monthly discharge simulations.

All in all, the here presented results should not be considered the final outcomes of a development towards hyper-resolution hydrological modelling. They are rather a first effort to map both opportunities and challenges of continuous simulations at 1 km resolution over large extents, with lessons learned for iterative open-ended model evaluation and subsequent improvement of model parameterization and configuration (Gleeson et al., 2021; Bierkens et al., 2015). As such, the results here should be seen as first steps in the realm of hyper-resolution with many more to follow. Despite the currently limited availability of observation dataset at matching spatial resolution and the still very demanding computations, we share the positive outlook of O'Neill et al. (2021) and are convinced that on-going data collection efforts, advanced satellite missions, and ever-powerful computers will eventually result in hyper-resolution becoming the default in the foreseeable future, and in turn providing globally continuous yet locally relevant results.

**Data and code availability.** Model output was evaluated using the 'pcrglobwb_utils' package (Hoch, 2021). As the original model output requires a lot of disk space, only the output from the evaluation step are provided under https://doi.org/10.5281/zenodo.6390219. For the original output, please contact the authors.

**Author contributions.** JMH, EHS, and MB designed and supervised the experiment. EHS prepared the model data and performed the analyses with critical feedback from all co-authors. JMH developed the code for model analysis and evaluated model output. JMH led the manuscript writing process with contributions from all co-authors.

**Competing interests.** The contact author has declared that neither they nor their co-authors have any competing interests.

**Acknowledgements.** JHM, ESH, and MB acknowledge funding from the Climate-KIC project "Agriculture Resilient, Inclusive, and Sustainable Enterprise (ARISE)". NW acknowledges funding from NWO 016.Veni.181.049. We furthermore thank SURF (www.surf.nl) for the support in using the National Supercomputer Snellius.



## A.     Appendices

### A1. Appendix A

This part provides some brief description about the PCR-GLOBWB model input used in this study. For more extensive details, we refer to previous PCR-GLOBWB model studies, particularly the ones at 50 km resolution
(van Beek et al., 2011; van Beek, 2008; van Beek and Bierkens, 2008) and 10 km resolution (Sutanudjaja et al., 2018) as well at 1 km resolution (Sutanudjaja et al., 2011).

### Meteorological forcing

As the meteorological forcing input, PCR-GLOBWB requires daily spatial fields of precipitation and temperature, as well as reference potential evaporation. For this study, reference potential evaporation is calculated using the
Penman-Monteith method following the FAO guidelines (Allen et al., 1998) that requires net radiation, wind speed, and vapor pressure deficit (see van Beek (2008) for details).

For all of the aforementioned required meteorological fields, we downloaded the ERA5-Land dataset (Muñoz-Sabater et al., 2021), which are provided at 6 arc-minute and hourly resolution. We then resampled the ERA5-Land precipitation and temperature variables, to daily spatial fields at 5 arc-minute resolution so that they can be
used for our 10 km model runs. We also resampled the required variables for the reference potential evaporation calculation at 5 arc-minute resolution and used it as the reference potential evaporation input of our 10 km model runs.

For our 50 km model run, we upscaled or averaged the 5 arc-minute daily precipitation, temperature, and reference potential evaporation variables to their 30 daily arc-minute fields. For 1 km model runs, we downscaled the daily
precipitation and temperature fields from 5 arc-minute to 30 arc-second using the monthly lapse rate fields from Sutanudjaja et al. (2018). We refer to Sutanudjaja et al. (2011) for more details about the downscaling procedure. The downscaled 30 arc-second temperature fields were also used to calculate the reference potential evaporation fields based on the Hamon-method (Hamon, 1963) that requires only daily mean temperature as its input. We then used these Hamon 30 arc-second fields to downscale the 5 arc-minute fields of the Penman-Monteith reference
potential evaporation to 30 arc-second resolution.

### Land surface: soil, and cover, and topography

For soil parameterization, the maps from SoilGrids250 (Hengl et al., 2017) were used in this study. This is different than previous PCR-GLOBWB studies that used the Digital Soil Map of the World (DSMW; FAO, 2007). An obvious advantage using SoilGrids250 from is its sufficiently fine resolution at 0.002 arc-degree resolution
(0.12 arcmin, about 250 m at the equator), while DSMW may not be suitable for model grids finer than 5 arc-minute resolution (see e.g. Batjes, 2012).

The PCR-GLOBWB model cannot directly use the soil information of SoilGrids250, which not only has a finer spatial resolution than 1 km PCR-GLOBWB, but also specifies only some general attributes such as soil texture. Here we transformed these attributes of SoilGrids into soil hydraulic properties, such as water holding capacity,
field capacity and wilting point, using the pedotransfer functions from Balland and Arp (2005) and Balland et al. (2008). These pedotransfer functions allow the estimation of bulk density and related soil-hydraulic properties at any given depth, which is required to link the layer information from SoilGrids to the two-layer schematization in PCR-GLOBWB. We derived these soil properties at the spatial resolution of 0.002 arc-degree and subsequently upscaled and resampled them to various PCR-GLOBWB model resolutions used in this study, i.e. 30 arc-second
(~1 km at the equator), 5 arc-minute (~10 km) and 30 arc-minute (~50 km).

For the standard parameterization of the land cover the following data sets were combined: the map of Global Land Cover Characteristics Database (GLCC) version 2.0 (Loveland et al., 2000) with the land cover classification following Olson (1994a, b) and the parameter sets from Hagemann et al. (1999) and Hagemann (2002). For the extent of irrigation areas, the map of Global Food Security-support Analysis Data (GFSAD) version 1.0
(Teluguntla et al., 2016) was used. The GLCC and GFSAD maps are available at 30 arc-second resolution (~1 km). Hence, for 1 km model runs, only one land cover type exists per cell. For the resolutions of 10 km and 50





km, land cover types were divided into four land cover types consisting of tall natural vegetation, short natural vegetation, non-paddy irrigated crops, and paddy irrigated crops (i.e. wet rice). Irrigation land cover types (i.e. paddy and non-paddy), including their crop calendars and growing season lengths, were parameterized based on the data set of MIRCA2000 (Portmann et al., 2010) and the Global Crop Water Model (Siebert and Döll, 2010).

Detailed description can be found in the key PCR-GLOBWB model description literature (Sutanudjaja et al., 2011, 2018; van Beek et al., 2011).

For topographical related input, we made use of the state of the art Multi-Error-Removed Improved-Terrain Hydro digital elevation (MERIT Hydro DEM, Yamazaki et al. (2019)) that is available at 3 arc-second resolution (~90 m at the equator). This is different than the previous PCR-GLOBWB studies (Sutanudjaja et al., 2018; van Beek

et al., 2011) that used previous generation DEM datasets, such as HydroSHEDS (Lehner et al., 2008) or GTOPO30 (Gesch et al., 1999). The 3 arc-second MERIT Hydro DEM was upscaled to the resolutions 30 arc-second (1 km), 5 arc-minute (10 km) and 30 arc-minute (50 km). Yet, we also used its original 3 arc-second resolution elevation values to obtain several sub-grid variability parameters that influences various schemes in PCR-GLOBWB, such as for runoff-infiltration partitioning, interflow, groundwater recharge and capillary rise, as well as evaporation

processes (van Beek, 2008; van Beek and Bierkens, 2008; Hagemann and Gates, 2003; Todini, 1996).

**Groundwater**

In PCR-GLOBWB, groundwater discharge (also commonly known as groundwater baseflow) depends on a linear storage-outflow relationship, in which a groundwater recession coefficient field is calculated following the drainage theory of Kraijenhoff Van de Leur (1958) based on the drainage network density and aquifer properties.

For the drainage density, we used the estimate from van Beek and Bierkens (2008). The aquifer properties were estimated from the GLobal HYdrogeology MaPS (GLHYMPS) dataset of Gleeson et al. (2014). These datasets are reasonably sufficiently fine for modeling at 1 km resolution and can be upscaled to 10 km resolution and 50 km resolution.

**Surface water routing: lakes, reservoirs and drainage/river network**

PCR-GLOBWB also includes lakes and reservoirs that are taken from the Global Lakes and Wetlands Database (GLWD) of Lehner and Döll (2004) and from the Global Reservoir and Dam Database (GRanD) of Lehner et al. (2011). We translated these datasets to 30 arc-second, 5 arc-minute and 30 arc-minute spatial fields of various lake and reservoir properties, such as surface areas and capacities. We combined these datasets to the drainage/river network maps, particularly to take into account the outlet positions of lakes and reservoirs.

For the drainage network maps, we made use of the existing 30 arc-minute and 5 arc-minute maps from Sutanudjaja et al. (2018), and the 30 arc-second one from the HydroSHEDS (Lehner et al., 2008). Note that the MERIT Hydro, which we used as the source of DEM, provides a drainage/river network at 3 arc-second resolution only and, therefore, cannot be used for this study.






## A2. Appendix B

Appendix B provides additional plots and analyses of model results.

In Figure B1, the improvement of KGE values per stations are mapped. To that end, it was assessed whether the KGE value for the 1k_1k run was higher than for either the 10k_10k run or the 50k_50k run, or higher for both runs, or not. Results indicate that KGE values improve at large, with only around 7 % of the stations showing no improvement at all when refining model resolution.

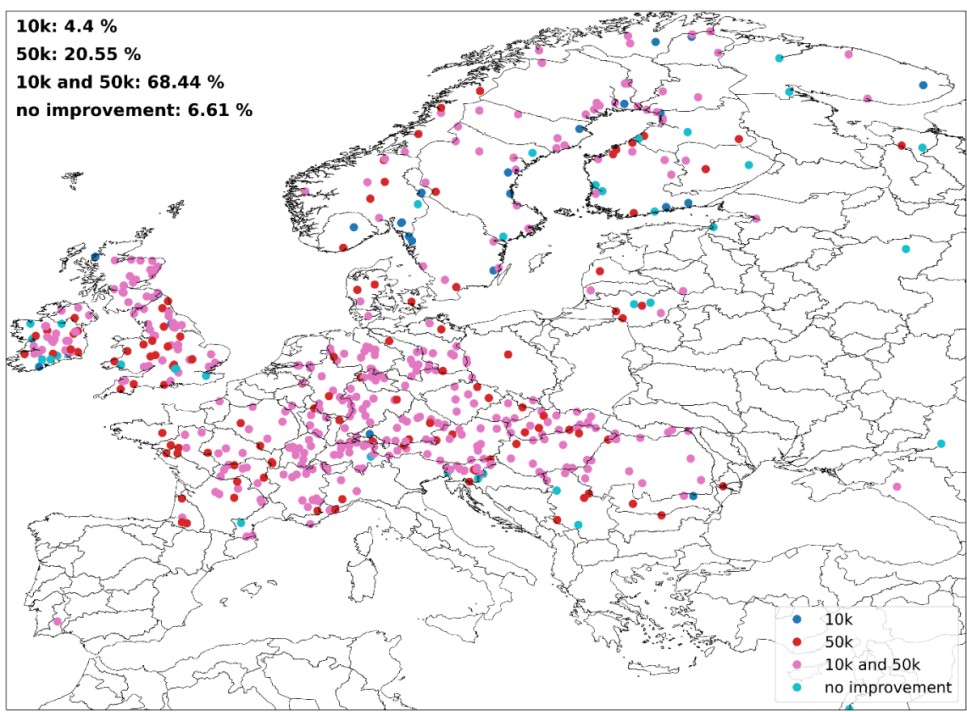

**Figure B1: Map of stations where the 1k run showed improved to only the 10k run, only the 50k run, both runs, or none of the runs.**




Figure B2 until Figure B4 provide scatter plots of $R^2$ and RRMSE values obtained by evaluating various model variables with observational datasets at the water province level. While panel A compares the impact of different model resolutions (1 km, 10 km, 50 km), panels B and C compare the impact of different forcing resolutions: B) 1 km model resolution with 1 km forcing resolution against 1 km model resolution and 10 km and 50 km forcing resolution, respectively; C) 10 km model resolution with 10 km forcing resolution against 10 km model resolution and 50 km forcing resolution,

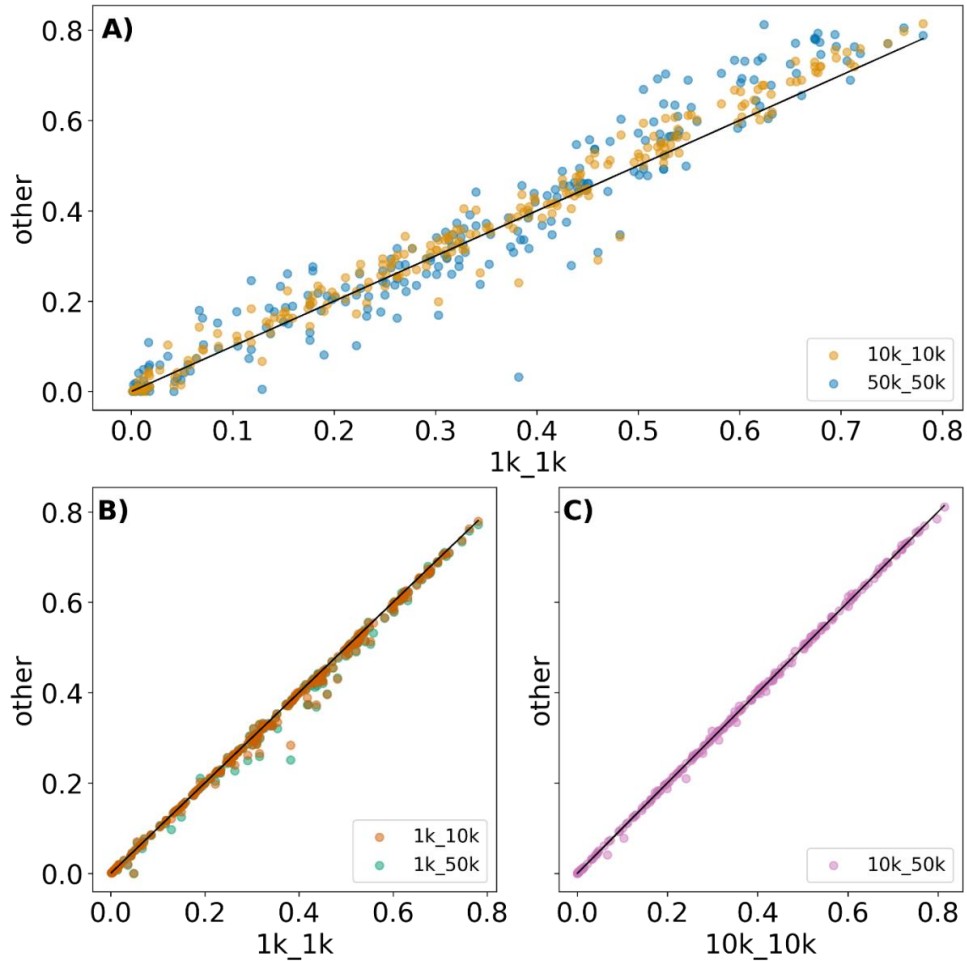

**Figure B2: Scatterplot of coefficient of determination ($R^2$) values obtained in evaluation of simulated upper soil moisture with ESA-CCI data for: A) comparing different model resolutions; B) comparing different forcing resolutions for 1k runs; C) comparing different forcing resolutions for 10k.**





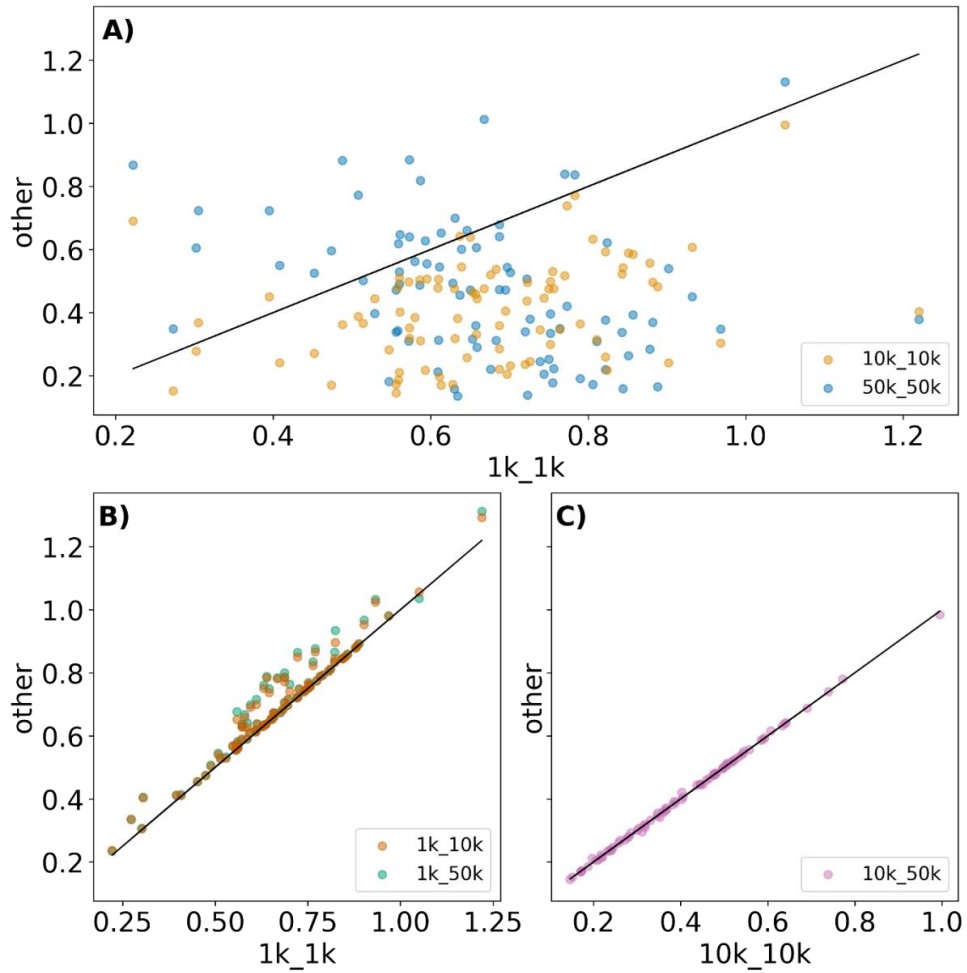

**Figure B3: Scatterplot of RRMSE values obtained in evaluation of simulated total evaporation with GLEAM data for: A) comparing different model resolutions; B) comparing different forcing resolutions for 1k runs; C) comparing different forcing resolutions for 10k runs.**




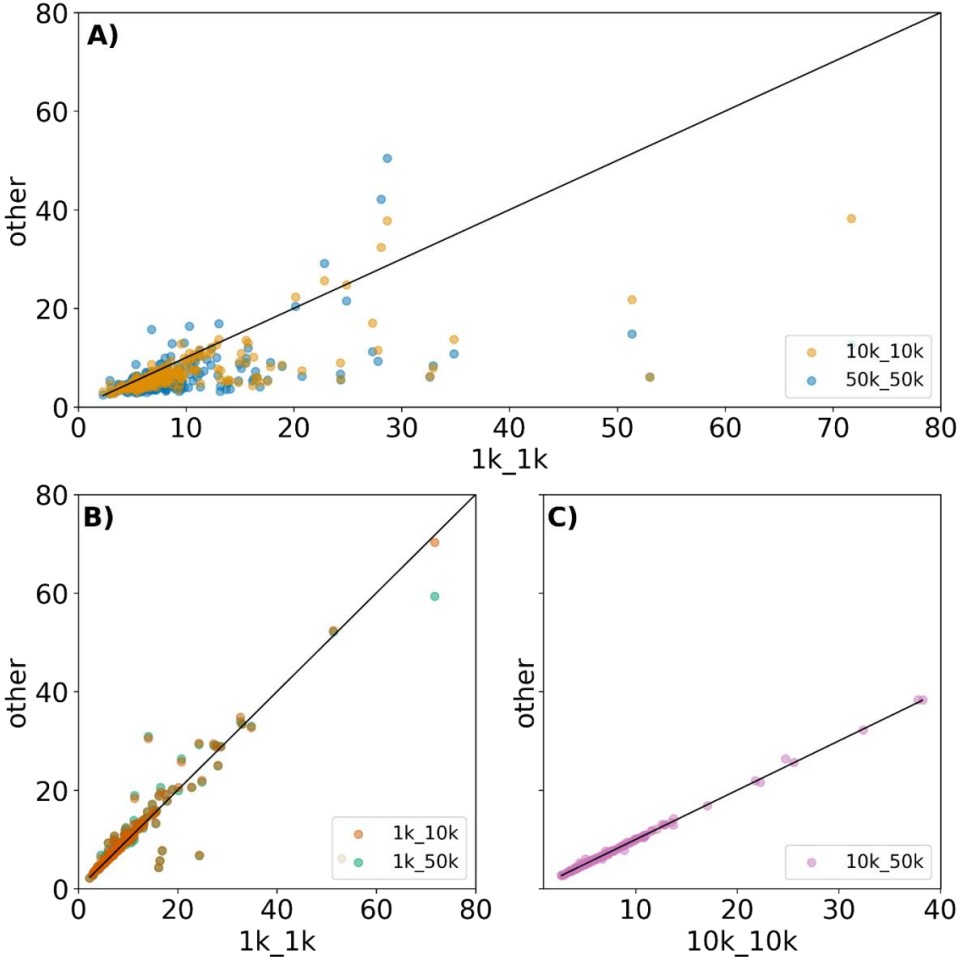

**Figure B4: Scatterplot of RRMSE values obtained in evaluation of simulated terrestrial water storage anomaly with GRACE/GRACE-FO data for: A) comparing different model resolutions; B) comparing different forcing resolutions for 1k runs; C) comparing different forcing resolutions for 10k runs.**



Figure B5 depicts the categorization of each water province conditional on their R2 and RRMSE values. Four categories were defined: with low and high R2 values and low and high RRMSE values. The median R2 respectively RRMSE value was used to define 'low' and 'high' values. It is shown that there is not really a consistent spatial patterns where which category dominates. Additionally, this figure shows the water provinces

5    with highest R2 and lowest RRMSE and vice versa as well as those provinces for which we analysed the impact of removing snow cover in the evaluation of TWS anomalies.

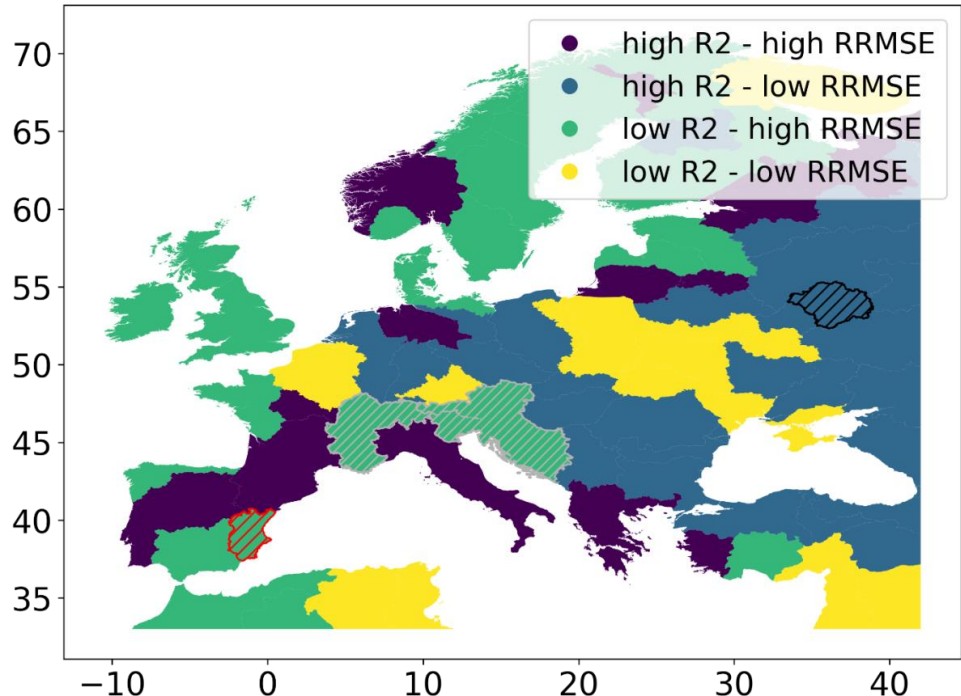

**Figure B5: Categorization of water provinces according to their R2 and RRMSE values. Red marker refers to province with lowest R2 value, black marker refers to province with highest R2 value (ID 204 and 224 in Figure 10, respectively).**
10   **Grey marker refers to water provinces for which snow cover correction of TWS anomaly were effective (see section 3.4).**



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
