# Peer review of "Hyper-resolution PCR-GLOBWB: opportunities and challenges of refining model spatial resolution to 1 km over the European continent"

_EGUsphere, 2022_

## Author Comment (AC1)

Dear Reviewer,

Many thanks for taking the time and reviewing the submitted manuscript. We also thank you for your kind words, and even more so your critical yet constructive evaluation.

Below, we address your specific comments (in blue) and will outline how we will improve a re-submitted manuscript, especially in the introduction and discussion sections.

What spatial extent constitutes a global hydrological model? Is the continental scale as given in the present study can be treated global? This question arises because of the statement that few studies have attempted hyper-resolution modelling over CONUS, but they do not have global coverage. In a strict sense, why can't this study be termed as a continental scale application?

Many thanks for this comment. A global hydrological model covers the entire terrestrial surface except for, at least in most instances, areas below/above +60/-60 degrees North. In doing so, it typically draws upon input datasets with global extent, a uniform parameterization, and is not calibrated regionally or locally, to facilitate comparability across the Globe. You rightly point out that our application is not 'global' as it covers only the European continent. However, our application merely employs global input datasets, a global parameter set (as it's been derived from coarser model versions), and is not calibrated regionally or locally, which is key in deriving potentially globally-transferrable findings from model evaluation. More generally speaking, one could say that a global hydrological model allows continental scale applications everywhere around the globe, but a continental model not for global applications (even though the numerical scheme could probably be fed with global data and be run if the model does not depend on locally calibrated parameter values).

In the revised version of the manuscript, **we will make it clearer that we our aim is to evaluate a global hydrological model in a continental scale application** as the 1 km version of PCR-GLOBWB is still very much under development and therefore its evaluation profits greatly from being applied to data-rich areas now (see also your other remark below). On the longer term, the model will of course be applied and evaluated globally.

The major issue in hyper-resolution modelling is modelling the physical processes happening at smaller scales. When developing a hyper-resolution model over large spatial extent, the physical processes to be considered would vary from region to region. How to account for the spatial variation in physical processes in the model? Can a generic model be applied over the entire continent/globe without accounting for region specific physical processes? Or how to develop model which can consider automatically, the various hydrological processes appropriate for a region within the model domain?

We kindly thank the reviewer for the critical comment. Indeed, representing physical processes happening at smaller scales is one of the "grand challenge" (Wood et al., 2011; Beven and Cloke, 2012). While for coarser spatial resolutions, it was deemed acceptable to subsume these processes in one or more parameters, this approach reaches its limits at hyper-resolution. To what extent it is possible to 'blindly' apply a single process representation with varying parameters at hyper-resolution (and how to move forward from there) is in fact one key questions of the submitted manuscript (see page 3, lines 5-6). While we already reflect on the answer to this question and their implications in the current version of manuscript various times (e.g.: page 1, line 24; page 15, line 36), **we will sharpen the focus on a revised version**.

Please note that it is a key feature of global hydrological models to use one uniform way of describing or parameterizing physical processes world-wide. Balancing generality and specificness determine hereby the overall accuracy of the model. The range of available literature on global hydrological modelling (see Bierkens (2015) for the latest state-of-the-art review) strongly indicates that such an approach is scientifically sound and that its results are of societal importance. In a revised version of the manuscript, **we will re-formulate the relevant sections such that the interplay between continental/regional specificity and global modelling approach becomes more evident**.

One reason that is often mentioned as an advantage of hyper-resolution modelling is the ability to simulate hydrological processes over data scarce regions. If data scarcity prevents us from developing a detailed model over a particular catchment, then how can be confident about the processes simulated by a global model over such data scarce regions? Further, in hydrology, studies are there to demonstrate the transfer the information obtained over a data-rich region to a data scarce region with similar characteristics. How global hyper-resolution modelling will add value to the existing methods in understanding the processes over data scarce regions?

This is an interesting question. We would argue that it is not the advantage of hyper-resolution modelling to be able to simulate hydrological processes over data-scarce areas. It is rather the advantage of global hydrological models in doing so, regardless the spatial resolution chosen. Nevertheless, we agree that global hydrological models should not be derived in data-scarce areas and then applied to data-rich areas, but vice versa. This is also why we opted for Europe, a data-rich region, as initial test case for the 1 km evaluation and starting point for model development efforts on the global scale. Once a global model is developed, its transferability to data-scarce areas can be tested. Based on the 10 km model, overall transferability of model skill is expected to be good (see Fig. 2a and Fig. 8a in Sutanudjaja et al. (2018) for a comparison of discharge and terrestrial water storage with observations). As such, the here presented study does not intend to improve modeling of data-scarce areas, but pave the way for improved representation of hydrological processes "everywhere and locally relevant" via hyper-resolution (Bierkens et al., 2015).

As the topic of knowledge transfer from data-rich (what we do in the submitted manuscript) to data-scarce areas (what we will do in future research) is of importance, as you rightly indicate, we will put extra emphasis in discussing this topic in revised version of the manuscript.

On a similar note, is it possible to develop a nested model structure that is followed in numerical weather modelling? i.e., develop coarse resolution model over large region and the outputs of this would act as boundary conditions of a nested model over a smaller spatial extent but at a much finer spatial resolution.

Thank you for your remark and great suggestion. Such nested model structures are not very common in global hydrological modelling, most likely because producing bespoke local high-detail models is a bit against the 'philosophy' of global hydrological modelling. What is instead done in instances where finer output is required than model resolution allows, is downscaling. There are, however, a few examples where nested modelling approaches are used, but then with the aim to include physical processes that are not represented by a GHM, such as detailed two-dimensional floodplain routing (Hoch et al., 2019) or coastal boundaries (Eilander et al., 2022). This is not to say that applying nested models is not viable, but it would require common efforts of global hydrological modelers and experts at the regional/catchment scale. Another option would be to use flexible meshes which can be refined where needed. Surely, both are avenues which hydrological studies should explore in more detail to advance the societal impact of hydrological models. At the moment, unfortunately, this will be only feasible for bespoke case studies and cannot be automated for any area of interest – which is, again, against the global transferability and comparability mindset of the global hydrological modelling approach.

Can the authors throw some light on the improvements to be made on the numerical aspect of the models? i.e., how to improve the efficiency of the models through novel and recent numerical schemes? This might save time during model runs.

We thank the reviewer for pointing this out. As we have mentioned in the manuscript already, we applied the same numerical scheme as used in the 10 km model (Sutanudjaja et al., 2018). Our study is hence a 'blind-test' how good this scheme would work when directly applied to a much finer spatial resolution. While the overall ability for hydrological simulations is provided, the current scheme is not suitable to execute lateral processes between cells efficiently, which has consequences for the model's ability to simulate groundwater and river discharge (even though our validation shows very satisfying performance over Europe), as we also point out in the manuscript (page 5, line 20) already. Possible ways forward are mentioned too (page 15, last paragraph; page 16, first paragraph and line 6), but **we will ensure that the role and limitations of the numerical scheme is discussed more clearly** in the revised manuscript. Also, **we will expand this section** and include additional relevant literature such as the LUE framework (de Jong et al., 2022), Distributed

memory parallel modeling (Verkaik et al., 2021), or the option for running models on GPUs (Shaw et al., 2021) or XPUs in general.

Why can't the 1K model be validated on a grid-by-grid basis using the available high quality in situ observations even for a smaller time period? For example, soil moisture and ET can be validated using in-situ datasets. Further, for soil moisture, comparison can be made against SMAP data that are available at relatively higher spatial resolutions than the ESA-CCI data? Similarly, for ET too, comparison can be made against available high-resolution products such as MODIS 16 ET, PML-V2 product (Zhang et al., 2019) etc. This can be useful to test if the model is really performing in a hyper-resolution manner. At present, I feel the model evaluation is not rigorous enough.

Many thanks for bringing this up. You are indeed right that a second layer of evaluations is needed, namely against high-resolution satellite or in-situ observations. Research is already on-going or planned for various catchments where such data is available. As the submitted manuscript is intended to serve as a baseline study of where the 1 km PCR-GLOBWB model stands and what current challenges are, we did not want to include this second layer as it would overload the manuscript.

Moreover, there is a practical limitation of using such high-resolution data (e.g. 500 m data) in the research framework of this study. As we evaluate three model resolution (1 km, 10 km, and 50 km) with observational data at 25 km and 300 km, a spatial aggregation level was needed which allows for a fair yet thorough comparison. Hence, we used water provinces over which we averaged results. The added value of very detailed observational data would thus be diluted by spatial averaging. Because of this, we do not claim but only hypothesize that the resolution of observational data impacts the outcome of the evaluation. **In the revised manuscript, we will lay out better the motivation and limitations of the use of water provinces** in our study.

Due to our main goal of having a very robust baseline study, we furthermore opted for observational data which has a long record rather than having highest spatial resolution available. We think that employing data from 2015 onwards (as in the case of SMAP data) does not include enough variability to allow for answering our research questions to our fullest satisfaction. For follow-up studies, especially those focusing on hyper-resolution data only and not on a wide range of spatial resolutions, this dataset can become key in the validation strategy. **In the revised manuscript, we will better highlight the need to compare against observations with sufficiently fine spatial resolution** in future research activities.

**References**

Beven, K. J. and Cloke, H. L.: Comment on "Hyperresolution global land surface modeling: Meeting a grand challenge for monitoring Earth's terrestrial water" by Eric F. Wood et al., Water Resour. Res., 48, https://doi.org/10.1029/2011WR010982, 2012.

Bierkens, M. F. P.: Global hydrology 2015: State, trends, and directions: Global Hydrology 2015, Water Resour. Res., 51, 4923–4947, https://doi.org/10.1002/2015WR017173, 2015.

Bierkens, M. F. P., Bell, V. A., Burek, P., Chaney, N., Condon, L. E., David, C. H., de Roo, A., Döll, P., Drost, N., Famiglietti, J. S., Flörke, M., Gochis, D. J., Houser, P., Hut, R., Keune, J., Kollet, S., Maxwell, R. M., Reager, J. T., Samaniego, L., Sudicky, E., Sutanudjaja, E. H., van de Giesen, N., Winsemius, H., and Wood, E. F.: Hyper-resolution global hydrological modelling: what is next?: "Everywhere and locally relevant," Hydrol. Process., 29, 310–320, https://doi.org/10.1002/hyp.10391, 2015.

Eilander, D., Couasnon, A., Leijnse, T., Ikeuchi, H., Yamazaki, D., Muis, S., Dullaart, J., Winsemius, H. C., and Ward, P. J.: A globally-applicable framework for compound flood hazard modeling, Hydrological Hazards, https://doi.org/10.5194/egusphere-2022-149, 2022.

Hoch, J. M., Eilander, D., Ikeuchi, H., Baart, F., and Winsemius, H. C.: Evaluating the impact of model complexity on flood wave propagation and inundation extent with a hydrologic–hydrodynamic model coupling framework, Nat. Hazards Earth Syst. Sci., 19, 1723–1735, https://doi.org/10.5194/nhess-19-1723-2019, 2019.

de Jong, K., Panja, D., Karssenberg, D., and van Kreveld, M.: Scalability and composability of flow accumulation algorithms based on asynchronous many-tasks, Comput. Geosci., 162, 105083, https://doi.org/10.1016/j.cageo.2022.105083, 2022.

Shaw, J., Kesserwani, G., Neal, J., Bates, P., and Sharifian, M. K.: LISFLOOD-FP 8.0: the new discontinuous Galerkin shallow-water solver for multi-core CPUs and GPUs, Geosci. Model Dev., 14, 3577–3602, https://doi.org/10.5194/gmd-14-3577-2021, 2021.

Sutanudjaja, E. H., van Beek, R., Wanders, N., Wada, Y., Bosmans, J. H. C., Drost, N., van der Ent, R. J., de Graaf, I. E. M., Hoch, J. M., de Jong, K., Karssenberg, D., López López, P., Peßenteiner, S., Schmitz, O., Straatsma, M. W., Vannametee, E., Wisser, D., and Bierkens, M. F. P.: PCR-GLOBWB 2: a 5 arcmin global hydrological and water resources model, Geosci. Model Dev., 11, 2429–2453, https://doi.org/10.5194/gmd-11-2429-2018, 2018.

Verkaik, J., Hughes, J. D., van Walsum, P. E. V., Oude Essink, G. H. P., Lin, H. X., and Bierkens, M. F. P.: Distributed memory parallel groundwater modeling for the Netherlands Hydrological Instrument, Environ. Model. Softw., 143, 105092, https://doi.org/10.1016/j.envsoft.2021.105092, 2021.

Wood, E. F., Roundy, J. K., Troy, T. J., van Beek, L. P. H., Bierkens, M. F. P., Blyth, E., de Roo, A., Döll, P., Ek, M., Famiglietti, J., Gochis, D., van de Giesen, N., Houser, P., Jaffé, P. R., Kollet, S., Lehner, B., Lettenmaier, D. P., Peters-Lidard, C., Sivapalan, M., Sheffield, J., Wade, A., and Whitehead, P.: Hyperresolution global land surface

modeling: Meeting a grand challenge for monitoring Earth's terrestrial water, Water Resour. Res., 47, https://doi.org/10.1029/2010WR010090, 2011.

---

## Author Comment (AC2)

Dear Reviewer,

Many thanks for your critical yet constructive review of the submitted manuscript. Before the manuscript is resubmitted, we will put additional emphasis on language and grammar.

On a more general note, and answering your question whether 'hyper-resolution is dead before it really started?', we would like to say that we do not find the presented results to be sobering, but very much meeting our expectations. Even though the topic of hyper-resolution hydrological modelling has now been discussed for a decade already, there have only been a few actual attempts to move hydrological models to a (sub-)kilometer resolution over large scales. As such, we think it would be illusive to assume that a first take (and that is exactly what the study presents) would yield major improvements across all evaluated variables, especially against the backdrop of observation data which indeed is not yet entirely commensurate as you rightly state. In a revised version of the manuscript, we will ensure that the 'expectation management' is improved with respect to what a first-generation hyper-resolution hydrological model can achieve.

Below, we address your further comments (in blue) and will outline how we plan to improve a revised manuscript.

In a way, the study has a conceptual problem, because upscaling and re-classification of soil texture and land cover (and water management/reservoirs?) was used to go from fine to coarse resolution. Thus the models are different not only in terms of spatial resolution and atmospheric forcing but also in terms of structure (i.e. different models at different resolution). Thus, comparability is not necessarily guaranteed, as claimed in the methods section. That's OK, but needs to be made transparent to the reader and discussed in detail. Perhaps it's one of the reasons why resolution does not do the trick in case of soil moisture and evaporation.

Many thanks for this comment. As also mentioned below, it is indeed true that model schematizations at different spatial resolutions are not identical, and that comparability is hampered. Schematizations of runs where only the forcing resolutions is changed are, however, identical. By using identical input data and parameters together with consistent scaling approaches, we aimed at minimizing differences across schematizations and their impact on comparability. In that sense, we do not compare model resolution in an isolated way, but rather model schematizations at different spatial resolutions (hyper-resolution and coarser) including their indirect impacts on how input data is processed internally. As these aspects are key, **we will add more transparency and explanation to the manuscript in general and especially its method section**.

The introduction is prominently missing a discussion of the recent relevant paper by Condon et al. (2022) on global (hpyer-resolution) groundwater modeling.

We kindly thank the reviewer for this literature suggestion. However, we could not find an article lead by (Laura) Condon from 2022 on hyper-resolution groundwater

modelling. We assume that this article from 2021 is meant instead, which indeed is a great addition to the manuscript and **will therefore be included in a revised version**.

2, 39: This statement is misleading. PFCONUS is just a naming convention (just as naming the setup of PCR-GLOBW over Europe PGEU). Of course ParFlow can be applied at the global scale, in principle; it's a generic simulation tool like many others.

Many thanks for pointing out this ambiguity. We are aware that both ParFlow and PCR-GLOBWB can be applied at any spatial resolution and spatial extent provided appropriate data is available to be fed into the model. What we failed to describe properly was that PFCONUS is a ParFlow model tailor-made for the CONUS region, whereas the 1k PGEU model – if you like – is using only data that could also be used for a global application. Being aware of this difference is crucial as more bespoke national or regional data sets will very likely be more accurate than data sets with global extend, which in turn will be reflected in the outcome of the evaluation. In a revised version of the manuscript, **we will rewrite the section under consideration such that this ambiguity is removed**.

4, 5: Here, additional information is required in the main text. From the appendix it follows that upscaling was used for soil texture and special classification for land cover was used to move from high to low resolution (how are reservoirs upscaled/downscaled?). Thus, the models are not identical in addition to the resolution of the forcing.

Thank you for your remark. Indeed, the model schematization are not identical across runs and resolutions. However, we never made this claim but only stated that the input data and parameters are identical. When creating schematizations at different spatial resolution, it cannot be avoided that these schematizations differ where data had to be down- or upscaled. Due to that, we did our best to keep the schematization as aligned as possible. In the revised manuscript, **we will put extra emphasis on explaining this properly**.

With respect to your specific questions on reservoirs, which we base on the GRanD data base (version 1.3) providing shapefiles of reservoirs, including information about their surface areas and capacities: for every resolution, we "rasterize" these shapefiles. In fact, it is more than just a "simple" rasterization process as we need to consider many factors, such as their locations to the drainage networks (at different resolutions), number of reservoirs within pixels, and so forth. If a pixel contains more than one reservoir (which is very likely in coarser resolution), we merged their surface areas and capacities and treated them as one reservoir. As such, the overall physical properties of reservoirs should not be overly different when moving from finer to coarser resolutions and thus their impact of flow estimates should be small. **We will append the appendix with this information and point towards it more prominently in the main body of the manuscript**.

Figure 3: remove 50k_50k from plot.

We thank the reviewer for proposing this improvement. While we initially kept the 50k_50k simulation in there as a reference point for the other simulations, we now agree that it does not add information to the plot and **will therefore be removed in the revised version of the manuscript**.

Why not applying the relative KGE to all variables (also soil moisture, ET)

Many thanks for this suggestion. Applying the KGE to all variables is not possible as particularly soil moisture evaluation needs to be treated carefully. Since satellite-based evaporation estimates are typically based on the first few centimeters of the top soil, PCR-GLOBWB uses the first 30 cm. Absolute values of soil moisture simulation and observations are therefore not directly comparable and we only can assess the correlation, as also mentioned in the manuscript on page 5. Consequently, we decided to use consistent metrics (RRMSE, R2) across all variables which we evaluated in space and time (i.e. soil moisture, evaporation, terrestrial water storage anomaly) and KGE for all variables which we only evaluated in time (i.e. discharge). Additionally, it is worth mentioning that by using the RRMSE we were able to account for differences in spatial variability of the signal we like to predict that exist between different areas, which is not possible when using KGE. Therefore, we are confident that the choice of metrics is well defined. Nevertheless, we may want to **add a better explanation of our reasoning to the revised manuscript**.

Figure 6: Replace "other" with correct information. Plot 1:1 line correctly everywhere. The plot almost suggests the 50k_50k is also doing better than 1k.

We thank the reviewer for pointing this out. **We will replace "other" with the actual run names** for improved comprehensibility. The 1:1 line in each plot, however, is plotted correctly, as is your observation that the 50k_50k run is (slightly) better than the 1k_1k run. This is also quantified in Table 3.

A couple of questions for the discussion and conclusions: Perhaps the observation data is not scale commensurate and can not be used to assess hyper-resolution modeling results? Perhaps PCR-GLOBWB is not scale commensurate and can not be used at hyper-resolution?

We thank the reviewer for these questions. In our opinion, it is rather the observation data that is not (yet) scale commensurate, at least when evaluating the simulations over a longer period (>5 years). The model itself should be applicable at the 1 km scale. Nevertheless, we need to acknowledge that there is still room for improvement which is not surprising as the study presents a real first-of-its-kind application and analysis of a hyper-resolution hydrological model at the continental scale. While we already discuss these questions in the manuscript, **we will ensure that they are answered more clearly in a revised version of the manuscript**.

---

## Author Response (AR1)

Dear Reviewers,

Many thanks for taking the time and reviewing the submitted manuscript. We also thank you for your kind words, and even more so your critical yet constructive evaluation. Based on your remarks, we think that the manuscript has improved greatly.

Below, we address your specific comments (in blue) and explain how and where we improve the revised manuscript. All sections that are updated or changed as result of the reviewer's comments are marked yellow in the revised manuscript.

In addition to answering and, where needed, implementing the reviewers' comments, we have put additional emphasis on language and grammar.

**Reviewer #1**

What spatial extent constitutes a global hydrological model? Is the continental scale as given in the present study can be treated global? This question arises because of the statement that few studies have attempted hyper-resolution modelling over CONUS, but they do not have global coverage. In a strict sense, why can't this study be termed as a continental scale application?

Many thanks for this comment. A global hydrological model covers the entire terrestrial surface except for, at least in most instances, areas below/above +60/-60 degrees North. In doing so, it typically draws upon input datasets with global extent, a uniform parameterization, and is not calibrated regionally or locally, to facilitate comparability across the Globe. You rightly point out that our application is not 'global' as it covers only the European continent. However, our application merely employs global input datasets, a global parameter set (as it's been derived from coarser model versions), and is not calibrated regionally or locally, which is key in deriving potentially globally-transferrable findings from model evaluation. More generally speaking, one could say that a global hydrological model allows continental scale applications everywhere around the globe, but a continental model not for global applications (even though the numerical scheme could probably be fed with global data and be run if the model does not depend on locally calibrated parameter values).

In the revised version of the manuscript, **we have made it clearer at multiple locations (abstract; p. 3, lines 19-21; p. 3, line 49; p. 15, line 7; p. 16, lines 36-39) that we our aim is to evaluate a global hydrological model in a continental scale application** only as the 1 km version of PCR-GLOBWB is still very much under development and therefore its evaluation profits greatly from being applied to data-rich areas now (see also your other remark below). On the longer term, the model will of course be applied and evaluated globally. **This aspect is now discussed in more detail too at the above-mentioned locations in the revised manuscript.**

The major issue in hyper-resolution modelling is modelling the physical processes happening at smaller scales. When developing a hyper-resolution model over large spatial extent, the physical processes to be considered would vary from region to region. How to account for the spatial variation in physical processes in the model? Can a generic model be applied over the entire continent/globe without accounting for region specific physical processes? Or how to develop model which can consider automatically, the various hydrological processes appropriate for a region within the model domain?

We kindly thank the reviewer for the critical comment. Indeed, representing physical processes happening at smaller scales is one of the "grand challenge" (Wood et al., 2011; Beven and Cloke, 2012). While for coarser spatial resolutions, it was deemed acceptable to subsume these processes in one or more parameters, this approach reaches its limits at hyper-resolution. To what extent it is possible to 'blindly' apply a single process representation with varying parameters at hyper-resolution (and how to move forward from there) is in fact one key questions of the submitted manuscript (see page 3, lines 5-6). While we already reflect on the answer to this question and their implications in the current version of manuscript various times (e.g.: page 1, line 24; page 15, line 36), **we have sharpened the focus in the revised version on page 3, lines 5-7, and page 16, lines 2-4.**

Please note that it is a key feature of global hydrological models to use one uniform way of describing or parameterizing physical processes world-wide. Balancing generality and specificness determine hereby the overall accuracy of the model. The range of available literature on global hydrological modelling (see Bierkens (2015) for the latest state-of-the-art review) strongly indicates that such an approach is scientifically sound and that its results are of societal importance. In a revised version of the manuscript, **we have re-formulate the relevant sections such that the interplay between continental/regional specificity and global modelling approach becomes more evident**. These changes go to a great extent hand in hand with the changes made based on the previous comment.

One reason that is often mentioned as an advantage of hyper-resolution modelling is the ability to simulate hydrological processes over data scarce regions. If data scarcity prevents us from developing a detailed model over a particular catchment, then how can be confident about the processes simulated by a global model over such data scarce regions? Further, in hydrology, studies are there to demonstrate the transfer the information obtained over a data-rich region to a data scarce region with similar characteristics. How global hyper-resolution modelling will add value to the existing methods in understanding the processes over data scarce regions?

This is an interesting question. We would argue that it is not the advantage of hyper-resolution modelling to be able to simulate hydrological processes over data-scarce areas. It is rather the advantage of global hydrological models in doing so, regardless

the spatial resolution chosen. Nevertheless, we agree that global hydrological models should not be derived in data-scarce areas and then applied to data-rich areas, but vice versa. This is also why we opted for Europe, a data-rich region, as initial test case for the 1 km evaluation and starting point for model development efforts on the global scale. Once a global model is developed, its transferability to data-scarce areas can be tested. Based on the 10 km model, overall transferability of model skill is expected to be good (see Fig. 2a and Fig. 8a in Sutanudjaja et al. (2018) for a comparison of discharge and terrestrial water storage with observations). As such, the here presented study does not intend to improve modeling of data-scarce areas, but pave the way for improved representation of hydrological processes "everywhere and locally relevant" via hyper-resolution (Bierkens et al., 2015).

As the topic of knowledge transfer from data-rich (what we do in the submitted manuscript) to data-scarce areas (what we will do in future research) is of importance, as you rightly indicate, **we have put this on the agenda for future research (p. 16, lines 39-41) in the revised manuscript.**

On a similar note, is it possible to develop a nested model structure that is followed in numerical weather modelling? i.e., develop coarse resolution model over large region and the outputs of this would act as boundary conditions of a nested model over a smaller spatial extent but at a much finer spatial resolution.

Thank you for your remark and great suggestion. Such nested model structures are not very common in global hydrological modelling, most likely because producing bespoke local high-detail models is a bit against the 'philosophy' of global hydrological modelling. What is instead done in instances where finer output is required than model resolution allows, is downscaling. There are, however, a few examples where nested modelling approaches are used, but then with the aim to include physical processes that are not represented by a GHM, such as detailed two-dimensional floodplain routing (Hoch et al., 2019) or coastal boundaries (Eilander et al., 2022). This is not to say that applying nested models is not viable, but it would require common efforts of global hydrological modelers and experts at the regional/catchment scale. Another option would be to use flexible meshes which can be refined where needed. Surely, both are avenues which hydrological studies should explore in more detail to advance the societal impact of hydrological models. At the moment, unfortunately, this will be only feasible for bespoke case studies and cannot be automated for any area of interest – which is, again, against the global transferability and comparability mindset of the global hydrological modelling approach. **Therefore, we have decided to not include a discussion on nested modelling in the revised manuscript**.

Can the authors throw some light on the improvements to be made on the numerical aspect of the models? i.e., how to improve the efficiency of the models through novel and recent numerical schemes? This might save time during model runs.

We thank the reviewer for pointing this out. As we have mentioned in the manuscript already, we applied the same numerical scheme as used in the 10 km model (Sutanudjaja et al., 2018). Our study is hence a 'blind-test' how good this scheme would work when directly applied to a much finer spatial resolution. While the overall ability for hydrological simulations is provided, the current scheme is not suitable to execute lateral processes between cells efficiently, which has consequences for the model's ability to simulate groundwater and river discharge (even though our validation shows very satisfying performance over Europe), as we also point out in the manuscript (page 5, line 20) already. Possible ways forward are mentioned too (page 15, last paragraph; page 16, first paragraph and line 6), but **now the role and limitations of the numerical scheme is discussed more clearly (p. 14, line 16; p. 16, lines 10-14)** in the revised manuscript. Also, **section 3.5 as well as the conclusion (p. 16, lines 14-17) have been expanded** by including additional literature such as the LUE framework (de Jong et al., 2022, 2021), distributed memory parallel modeling (Verkaik et al., 2021, 2022), or the option for running models on GPUs (Shaw et al., 2021) or XPUs in general. On a more general note, the need for fit-for-purpose cyberinfrastructure as discussed by Condon et al. (2021) is discussed.

Why can't the 1K model be validated on a grid-by-grid basis using the available high quality in situ observations even for a smaller time period? For example, soil moisture and ET can be validated using in-situ datasets. Further, for soil moisture, comparison can be made against SMAP data that are available at relatively higher spatial resolutions than the ESA-CCI data? Similarly, for ET too, comparison can be made against available high-resolution products such as MODIS 16 ET, PML-V2 product (Zhang et al., 2019) etc. This can be useful to test if the model is really performing in a hyper-resolution manner. At present, I feel the model evaluation is not rigorous enough.

Many thanks for bringing this up. You are indeed right that a second layer of evaluations is needed, namely against high-resolution satellite or in-situ observations. **This is now clearly recommended on page 16, line 38**. Research is already on-going or planned for various catchments where such data is available. As the submitted manuscript is intended to serve as a baseline study of where the 1 km PCR-GLOBWB model stands and what current challenges are, we did not want to include this second layer as it would overload the manuscript.

Moreover, there is a practical limitation of using such high-resolution data (e.g. 500 m data) in the research framework of this study. As we evaluate three model resolution (1 km, 10 km, and 50 km) with observational data at 25 km and 300 km, a spatial aggregation level was needed which allows for a fair yet thorough comparison. Hence, we used water provinces over which we averaged results. The added value of very detailed observational data would thus be diluted by spatial averaging. It is only when merely evaluating hyper-resolution output that a gid-to-grid analysis becomes meaningful. Because of this, we do not claim but only hypothesize that the resolution

of observational data impacts the outcome of the evaluation. **In the revised manuscript, we have laid out better the motivation and limitations of the use of water provinces in our study (p. 4, lines 32-35) and recommend a grid-by-grid evaluation in a follow-up study when evaluating hyper-resolution output only (p. 15, lines 43-45).**

Due to our main goal of having a very robust baseline study, we furthermore opted for observational data which has a long record rather than having highest spatial resolution available. We think that employing data from 2015 onwards (as in the case of SMAP data) does not include enough variability to allow for answering our research questions to our fullest satisfaction. For follow-up studies, especially those focusing on hyper-resolution data only and not on a wide range of spatial resolutions, this dataset can become key in the validation strategy. **In the revised manuscript, we have motivated our decision more clearly (p. 4, lines 25-28) and have better highlighted the need to compare against observations with sufficiently fine spatial resolution in future research activities (p. 15, lines 39-40; p. 16, line 38).**

**Reviewer #2**

In a way, the study has a conceptual problem, because upscaling and re-classification of soil texture and land cover (and water management/reservoirs?) was used to go from fine to coarse resolution. Thus the models are different not only in terms of spatial resolution and atmospheric forcing but also in terms of structure (i.e. different models at different resolution). Thus, comparability is not necessarily guaranteed, as claimed in the methods section. That's OK, but needs to be made transparent to the reader and discussed in detail. Perhaps it's one of the reasons why resolution does not do the trick in case of soil moisture and evaporation.

Many thanks for this comment. As also mentioned below, it is indeed true that model schematizations at different spatial resolutions are not identical, and that comparability is hampered. Schematizations of runs where only the forcing resolutions is changed are, however, identical. By using identical input data and parameters together with consistent scaling approaches, we aimed at minimizing differences across schematizations and their impact on comparability. In that sense, we do not compare model resolution in an isolated way, but rather model schematizations at different spatial resolutions (hyper-resolution and coarser) including their indirect impacts on how input data is processed internally. As these aspects are key, **we have added more transparency and explanation to the revised manuscript (p. 3, 5-7; p. 4, 14-18; p. 15, 23; p. 16, 2-4)**.

The introduction is prominently missing a discussion of the recent relevant paper by Condon et al. (2022) on global (hpyer-resolution) groundwater modeling.

We kindly thank the reviewer for this literature suggestion. However, we could not find an article lead by (Laura) Condon from 2022 on hyper-resolution groundwater

modelling. We assume that this article from 2021 is meant instead, which indeed is a great addition to the manuscript and **its vision on the need for fit-for-purpose cyberinfrastructure has been added to the revised manuscript (p. 16, line 17)**.

2, 39: This statement is misleading. PFCONUS is just a naming convention (just as naming the setup of PCR-GLOBW over Europe PGEU). Of course ParFlow can be applied at the global scale, in principle; it's a generic simulation tool like many others.

Many thanks for pointing out this ambiguity. We are aware that both ParFlow and PCR-GLOBWB can be applied at any spatial resolution and spatial extent provided appropriate data is available to be fed into the model. What we failed to describe properly was that PFCONUS is a ParFlow model tailor-made for the CONUS region, whereas the 1k PGEU model – if you like – is using only data that could also be used for a global application. Being aware of this difference is crucial as more bespoke national or regional data sets will very likely be more accurate than data sets with global extend, which in turn will be reflected in the outcome of the evaluation. In a revised version of the manuscript, **we have removed the misleading statement and have rewritten the introduction such that this ambiguity is removed**. **Any discussion building upon this has been updated, such as p. 3, lines 18-19.**

4, 5: Here, additional information is required in the main text. From the appendix it follows that upscaling was used for soil texture and special classification for land cover was used to move from high to low resolution (how are reservoirs upscaled/downscaled?). Thus, the models are not identical in addition to the resolution of the forcing.

Thank you for your remark. Indeed, the model schematization are not identical across runs and resolutions. However, we never made this claim but only stated that the input data and parameters are identical. When creating schematizations at different spatial resolution, it cannot be avoided that these schematizations differ where data had to be down- or upscaled. Due to that, we did our best to keep the schematization as aligned as possible. In the revised manuscript, **we have put extra emphasis on explaining this transparently (p. 3, 5-7; p. 4, 14-18; p. 15, 23; p. 16, 2-4)**.

With respect to your specific questions on reservoirs, which we base on the GRanD data base (version 1.3) providing shapefiles of reservoirs, including information about their surface areas and capacities: for every resolution, we "rasterize" these shapefiles. In fact, it is more than just a "simple" rasterization process as we need to consider many factors, such as their locations to the drainage networks (at different resolutions), number of reservoirs within pixels, and so forth. If a pixel contains more than one reservoir (which is very likely in coarser resolution), we merged their surface areas and capacities and treated them as one reservoir. As such, the overall physical properties of reservoirs should not be overly different when moving from finer to coarser resolutions and thus their impact of flow estimates should be small. **We have**

**added a more elaborated explanation how reservoirs are upscaled/downscaled the appendix (p. 19).**

Figure 3: remove 50k_50k from plot.

We thank the reviewer for proposing this improvement. While we initially kept the 50k_50k simulation in there as a reference point for the other simulations, we now agree that it does not add information to the plot and **therefore have removed it in the revised version of the manuscript**.

Why not applying the relative KGE to all variables (also soil moisture, ET)

Many thanks for this suggestion. Applying the KGE to all variables is not possible as particularly soil moisture evaluation needs to be treated carefully. Since satellite-based evaporation estimates are typically based on the first few centimeters of the top soil, PCR-GLOBWB uses the first 30 cm. Absolute values of soil moisture simulation and observations are therefore not directly comparable and we only can assess the correlation, as also mentioned in the manuscript on page 5. Consequently, we decided to use consistent metrics (RRMSE, R2) across all variables which we evaluated in space and time (i.e. soil moisture, evaporation, terrestrial water storage anomaly) and KGE for all variables which we only evaluated in time (i.e. discharge). Additionally, it is worth mentioning that by using the RRMSE we were able to account for differences in spatial variability of the signal we like to predict that exist between different areas, which is not possible when using KGE. Therefore, we are confident that the choice of metrics is well defined. Nevertheless, **we have added a brief explanation of to the revised manuscript why using the KGE across all variables is not possible (p. 5, line 44)**.

Figure 6: Replace "other" with correct information. Plot 1:1 line correctly everywhere. The plot almost suggests the 50k_50k is also doing better than 1k.

We thank the reviewer for pointing this out. **We have replaced "other" with the actual run names** for improved comprehensibility. The 1:1 line in each plot, however, is plotted correctly, as is your observation that the 50k_50k run is (slightly) better than the 1k_1k run. This is also quantified in Table 3.

A couple of questions for the discussion and conclusions: Perhaps the observation data is not scale commensurate and can not be used to assess hyper-resolution modeling results? Perhaps PCR-GLOBWB is not scale commensurate and can not be used at hyper-resolution?

We thank the reviewer for these questions. In our opinion, it is rather the observation data that is not (yet) scale commensurate, at least when evaluating the simulations over a longer period (>5 years). The model itself should be applicable at the 1 km scale. Nevertheless, we need to acknowledge that there is still room for improvement

which is not surprising as the study presents a real first-of-its-kind application and analysis of a hyper-resolution hydrological model at the continental scale. While we already discuss these questions in the manuscript, **we have included additional discussions about the role of model and observation resolution in the revised manuscript (p.15, 25; p. 15, 39-40)**.

**References**

[revised manuscript text omitted]

---

## Author Response (AR2)

Dear reviewer, dear editor,

We kindly thank you for your critical and productive comments on the revised version of the manuscript. Based on these, we have restructured the manuscript and reformulated specific sections. Specifically, we have made the subsequent changes to the manuscript:

- We have formulated the objective more clearly in the introduction, which we then feature more prominently in the conclusions chapter as well as in the abstract. This way, the overall aim of the paper is clearer and less intertwined with other findings and aspects of the manuscript.
- As suggested, we have created a new chapter 4 which discusses the following three major points: model parameterization, scale commensurability, and computer requirements. This chapter has been complemented with content from the results chapter and the conclusions chapter. In this step, some less relevant or repetitive content was removed. Subsequently, the discussion was extended with new material focusing on the three covered topics.
- As a consequence of the previous two changes, the conclusions and recommendations chapter is now less lengthy and fuzzy but focusses on the main take-away messages.
- In the discussions and conclusions chapters, the topic of scale (in)commensurability of model and observational data has been addressed in more detail, and how it affects the ability to (in)validate the results and modelling approach. As this topic deserves a study itself, we tried to strike a balance between providing sufficient space and depth for this topic without overly extending the manuscript.
- Furthermore, we formulated conclusions in a more balanced way, that is by avoiding general formulations but stressing dependencies with the modelling approach as well as input and observational data we opted for in this study.
- Some very minor cosmetic changes have been made to the manuscript to improve readability and flow across chapters, especially with the newly created discussion chapter.

After having implemented your feedback and suggestions, we feel that the manuscript has improved further and now captures the opportunities and challenges alongside their associated uncertainties better and more concisely.